# Immunopeptidomic MHC-I profiling and immunogenicity testing identifies Tcj2 as a new Chagas disease mRNA vaccine candidate

Leroy Versteeg[1,2]*, Rakesh Adhikari[1], Gonteria Robinson[1], Jungsoon Lee[1], Junfei Wei[1], Nelufa Islam[1], Brian Keegan[1], William K. Russell[3], Cristina Poveda[1], Maria Jose Villar[1], Kathryn Jones[1,4], Maria Elena Bottazzi[1,4,5], Peter Hotez[1,4,5], Edwin Tijhaar[2ᵒ]*, Jeroen Pollet[1ᵒ]*

1 Texas Children's Hospital Center for Vaccine Development, Department of Pediatrics, Division of Tropical Medicine, Baylor College of Medicine, Houston, Texas, United States of America, 2 Cell Biology and Immunology Group, Wageningen University & Research, Wageningen, The Netherlands, 3 University of Texas Medical Branch, Mass Spectrometry Facility, UTMB Health, Galveston, Texas, United States of America, 4 Department of Molecular Virology and Microbiology, Baylor College of Medicine, Houston, Texas, United States of America, 5 Department of Biology, Baylor University, Waco, Texas, United States of America

ᵒ These authors contributed equally to this work.
* Leroy.Versteeg@bcm.edu (LV); Edwin.Tijhaar@wur.nl (ET); Jeroen.Pollet@bcm.edu (JP)

**Data Availability Statement:** Mass spec data available on Dryad.com: Title: Immunopeptidomics

## Abstract

*Trypanosoma cruzi* is a protozoan parasite that causes Chagas disease. Globally 6 to 7 million people are infected by this parasite of which 20–30% will progress to develop Chronic Chagasic Cardiomyopathy (CCC). Despite its high disease burden, no clinically approved vaccine exists for the prevention or treatment of CCC. Developing vaccines that can stimulate *T. cruzi*-specific CD8+ cytotoxic T cells and eliminate infected cells requires targeting parasitic antigens presented on major histocompatibility complex-I (MHC-I) molecules. We utilized mass spectrometry-based immunopeptidomics to investigate which parasitic peptides are displayed on MHC-I of *T. cruzi* infected cells. Through duplicate experiments, we identified an array of unique peptides that could be traced back to 17 distinct *T. cruzi* proteins. Notably, six peptides were derived from Tcj2, a trypanosome chaperone protein and member of the DnaJ (heat shock protein 40) family, showcasing its potential as a viable candidate vaccine antigen with cytotoxic T cell inducing capacity. Upon testing Tcj2 as an mRNA vaccine candidate in mice, we observed a strong memory cytotoxic CD8+ T cell response along with a Th1-skewed humoral antibody response. *In vitro* co-cultures of *T. cruzi* infected cells with splenocytes of Tcj2-immunized mice restricted the replication of *T. cruzi*, demonstrating the protective potential of Tcj2 as a vaccine target. Moreover, antisera from Tcj2-vaccinated mice displayed no cross-reactivity with DnaJ in lysates from mouse and human indicating a decreased likelihood of triggering autoimmune reactions. Our findings highlight how immunopeptidomics can identify new vaccine targets for Chagas disease, with Tcj2 emerging as a promising new mRNA vaccine candidate.

on T. cruzi – infected MC57G murine fibroblasts. DOI: 10.5061/dryad.3r2280gnx.

**Funding:** This research was supported by funding from the Southern Star Medical Research Institute and intramural funds from Texas Children's Hospital Center for Vaccine Development at Baylor College of Medicine. The funders had no role in study design, data collection and analysis, decision to publish, or preparation of the manuscript.

**Competing interests:** LV, RA, JL, JW, NI, BK, MJV, CP, KJ, MEB, PH and JP collaborated in the development of Tc24-C4, a vaccine candidate against Chagas Disease that is currently undergoing clinical evaluation. JP, MEB and PH are listed among the inventors on a Chagas disease vaccine patent, submitted by Baylor College of Medicine.

## Author summary

Chagas disease, caused by the parasite *Trypanosoma cruzi*, affects millions of people worldwide and leaves a substantial proportion of patients at risk for developing Chronic Chagasic Cardiomyopathy (CCC). Despite the disease's impact, no approved immunotherapies or vaccines currently exist. This study employs an advanced technique called "immunopeptidomics" to identify potential vaccine targets that can induce CD8+ cytotoxic T cells, able to detect and kill infected cells in *T. cruzi*–infected tissues. By analyzing infected cells, we identified *T. cruzi*–specific peptides displayed on these cells' surfaces. Notably, multiple peptides arose from Tcj2, one of five *T. cruzi-* DnaJ proteins. Employing an mRNA vaccine, this novel antigen was administered to mice, prompting a robust immune response, including strongly elevated levels of *T. cruzi*-specific cytotoxic T cells. Overall, this study highlights the strength of immunopeptidomics in parasitology, revealing a promising vaccine target for Chagas disease.

## Introduction

Chagas disease is a neglected tropical disease caused by the protozoan parasite *Trypanosoma cruzi*. Approximately 6–7 million people worldwide are affected by the disease, resulting in 10,000–50,000 deaths per year, and an estimated 65–100 million are at risk of contracting the infection [1,2]. After the acute stage of the infection, generally with flu-like symptoms, patients can develop chronic disease in 30–40% of cases, and 20–30% of all chronic cases involving Chagasic cardiomyopathy (CCC), characterized by arrhythmias, heart aneurysms or failure, stroke, megacolon or megaesophagus [3,4]. Chagas disease is endemic in 21 Latin American countries, where it has been confined to rural and poor areas where transmission by the Triatoma (kissing bug) vector is the main route of infection [4]. However, due to transmission of the disease through blood transfusion, organ transplantation or by congenital transmission, Chagas disease is a growing concern for non-endemic countries in North America, Europe, Australia, and Asia that import cases through migration and globalization [3,5]. Furthermore, it is estimated that the global economic burden of Chagas disease is almost $7 billion, exceeding that of global diseases like rotavirus and cervical cancer [6]. While the anti-trypanosomal drugs benznidazole and nifurtimox are approved for use, they have poor efficacy in the chronic phase and significant side effects [7]. Much effort has been made to find new solutions to cure Chagas disease, such as vaccines and other immunotherapies, as well as better drugs [8,9].

During *T. cruzi* infection, extracellular trypomastigotes penetrate host cells, subsequently infiltrate the phagolysosome and eventually enter the host cell cytoplasm (Fig 1). Here, they proliferate and evade the humoral immune response. During this intracellular stage, parasite-derived proteins that are discarded or secreted are processed by the proteasome and presented on MHC-I. This MHC-I presentation at the cell surface is crucial as it enables antigen specific cytotoxic CD8+ T cells (CTLs) to recognize, and subsequently kill, the *T. cruzi* infected cells. These antigen specific CTLs play an essential role in cell-mediated immunity and the control of *T. cruzi* infections [10,11]. Importantly, it is thought that most trypomastigote and amastigote proteins are not directly available for MHC-I presentation, because they are still contained within the parasite, therefore not accessible for processing by the proteasome localized in the cytoplasm. Knowing which *T. cruzi* peptides are displayed on MHC-I of infected cells gives crucial insights for selecting appropriate candidate vaccine antigens that can induce cytotoxic T cell able to kill *T. cruzi* infected cells.

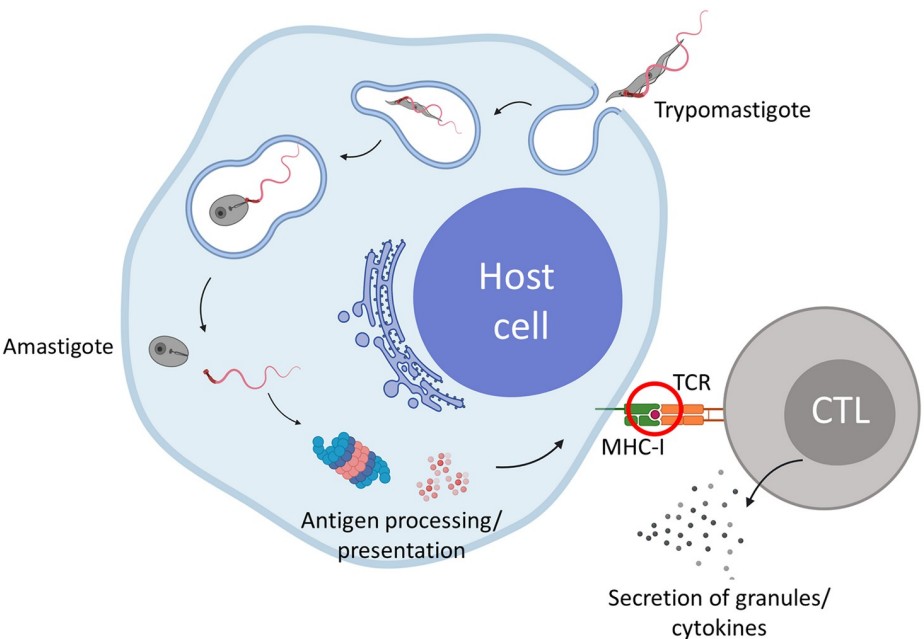

**Fig 1. Immunopeptodomics identifies which peptides derived from *T. cruzi* antigens can be detected by CD8 + cytotoxic T cells (CTLs).** After a *T. cruzi* trypomastigote invades a host cell, it enters the cytosol and transforms into the amastigote stage. During this process, several parasitic proteins become available for antigen processing by the proteasome followed by peptide presentation on MHC-I. This presentation can activate CTLs which secrete granules and cytokines to clear the infected cells. Immunopeptidomics will help us understand which proteins become available for the antigen processing and presentation machinery. Red encircled is the key event where immunopeptidomics is used to learn information on which peptides are presented. Created with Biorender.com.

Immunopeptidomics is the science that studies the peptides that are presented on major histocompatibility complex (MHC) proteins on the surface of cells (red circle, Fig 1). Within the last decade, major advances in the field of liquid chromatography–mass spectrometry (LC-MS/MS) have led to a significant increase in sensitivity and accuracy of data acquisition and prediction, which now allow the identification of MHC presented peptides at a superior sensitivity [12,13]. The repertoire of MHC-presented peptides, or so-called immunopeptidome, contains information of the health state of cells and is continuously surveilled by T cells [14,15]. Importantly, when a pathogen infects cells, pathogen-derived peptides will be presented by MHC-I [16]. The identification of pathogen-derived peptides presented on MHC by immunopeptidomics has already been done numerous times for various viruses [17–20], protozoans [12,21–23] and bacteria [24–27]. The results of these studies give insights into which pathogen derived peptides are presented on the surface of infected cells, thereby identifying the proteins that are accessible to the MHC-I presentation pathway. These proteins will be prime targets for inducing antigen specific CTLs that can kill infected target cells. Importantly, identification of peptides presented by *T. cruzi* infected cells has to this date not been performed. Thus, in combination with genomics and machine learning algorithms, immunopeptidomics offers new and innovative reverse vaccinology opportunities to address Chagas disease.

In recent years, mRNA vaccines have undergone significant development and are recognized as potent inducers of cell-mediated immunity, including MHC-I restricted CTLs [28,29]. The advantages of this vaccine platform for anti-parasitic vaccines have been

previously described and include the option of multivalency, rapid development and production, stability and strong induction of CTLs [30–32]. Promising results of mRNA vaccines for parasitic diseases have already been reported for malaria, toxoplasmosis and leishmaniasis [33–35]. The use of mRNA vaccines for Chagas disease has been proposed, and recently we published our work characterizing an mRNA vaccine construct encoding the flagellar calcium-binding protein Tc24 from *T. cruzi* [30,36–38].

Here, we report on the identification of a novel mRNA vaccine target for Chagas disease using immunopeptidomics. Through the analysis of the immunopeptidome of *T. cruzi* infected cells, using mass-spectrometry-based immunopeptidomics, we identified 24 unique *T. cruzi* peptides, derived from 17 different *T. cruzi* proteins. Of these proteins, Tcj2 –a known *T. cruzi* chaperone protein and part of the DnaJ (heat shock protein 40) family—was identified as a primary candidate vaccine antigen, since multiple unique Tcj2-derived peptides were identified in repeat experiments. Therefore, a Tcj2 mRNA vaccine was developed, and its immunogenicity tested *in vitro* and *in vivo* using a mouse model. The results revealed that when Tcj2 mRNA was formulated in lipid nanoparticles (LNPs), it provoked a robust immune response, displaying characteristics considered critical for a successful Chagas disease vaccine. Overall, our findings underscore the potential of immunopeptidomics in the identification of vaccine candidate antigens for parasitic diseases and highlights Tcj2-encoding mRNA formulated in LNPs as a promising vaccine candidate for Chagas disease.

## Results

### Analysis of peptides presented on MHC-I of *T. cruzi*—Infected cells

To analyze which proteins from *T. cruzi* are subject to antigen processing followed by presentation on MHC-I, the immunopeptidome of *T. cruzi*—infected cells was analyzed as shown in Fig 2A. Murine MC57G fibroblasts were infected with *T. cruzi* Tulahuen trypomastigotes for 48 hours, followed by lysing the infected fibroblasts and isolation of the MHC-I–peptide complexes using an immunoaffinity column that consisted of the anti-mouse MHC-I mAb (clone M1/42), covalently coupled to cross-linked agarose resin. Flow cytometry analysis showed increased binding of the M1/42 mAb to mouse MHC-I (H-2) expressing MC75G cells after 48-hour infection with *T. cruzi* compared to uninfected controls, suggesting a significant upregulation of MHC-I expression due to infection (Fig 2B).

Two immunopeptidomics experiments (technical replicates) were performed using lysate from *T. cruzi*—infected fibroblasts. Following the analysis of the peptides by LC-MS/MS, peptides were identified using available protein FASTA databases from *T. cruzi* and mouse. To further increase the specificity of the results, only peptide lengths between 8 and 15 amino acids (aa) were selected since these lengths have shown to bind most likely to MHC-I (H2-K$^b$ and H2-D$^b$) [39]. In the first experiment (#1) a total of 6 *T. cruzi* peptides were identified, as well as 1191 murine peptides (Fig 2C). When the experiment was repeated (experiment #2), 20 and 1475 peptides were identified from *T. cruzi* and mouse, respectively. When the immunopeptidome of non-infected fibroblasts was analyzed, 619 peptides from mouse and no peptides from *T. cruzi* were identified, indicating that mouse peptides are not mis-identified as *T. cruzi* peptides.

The length distribution of the identified peptides is displayed (Fig 2D). For *T. cruzi* peptides, the observed length was equally distributed between 8–14 aa with an average of 11.3. Interestingly, for murine self-peptides approximately 40% (experiment #1 39.5%, experiment #2 39.9%) of the peptides 8–15 aa in length were 8 or 9 aa long. In contrast, murine self-peptide length from non-infected fibroblasts showed a much higher number of peptides around 8 and

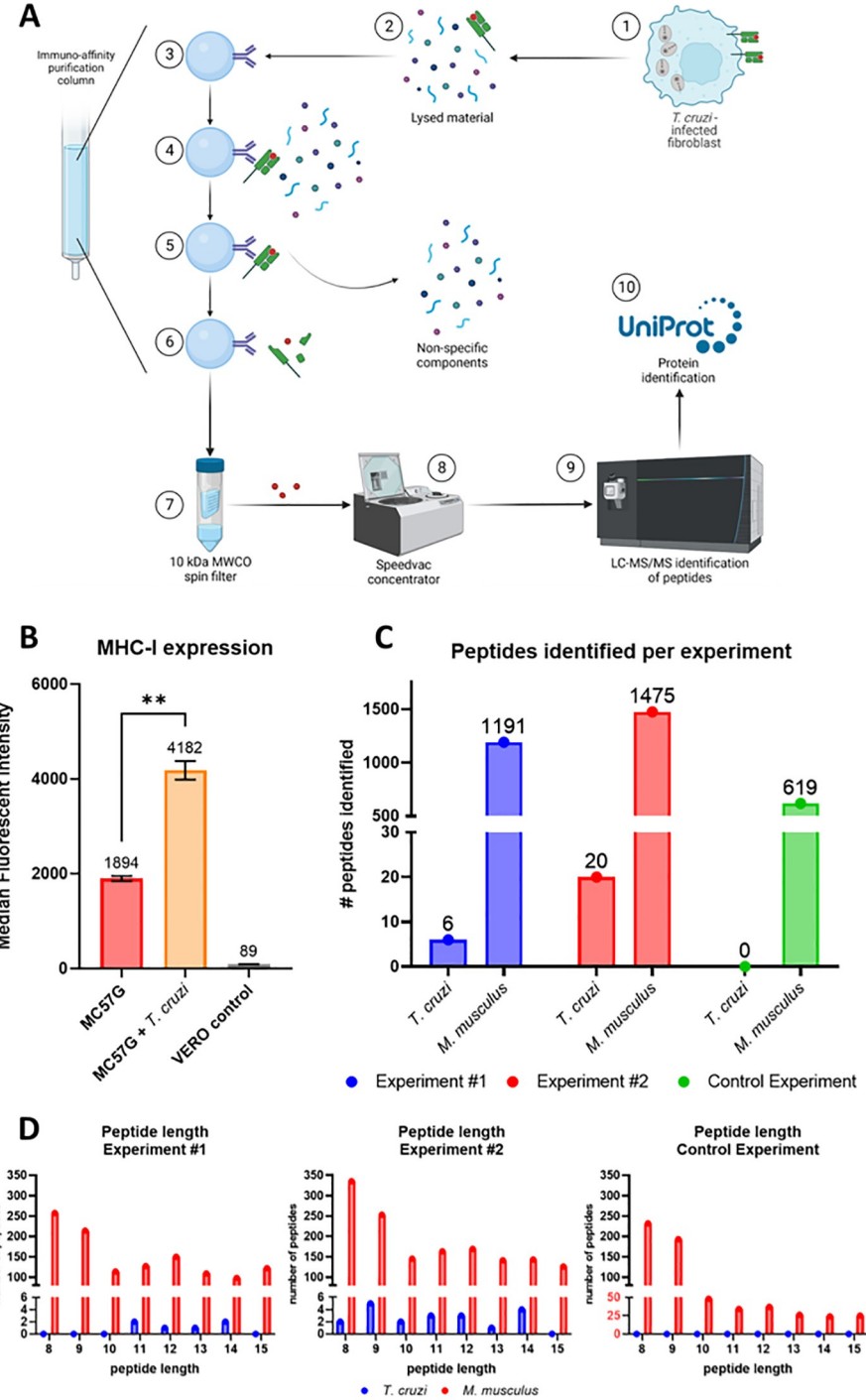

**Fig 2. Isolation and identification of the peptides presented on MHC-I of *T. cruzi* infected MC57G fibroblast cells.**
A) Schematic overview of the immunopeptidomics workflow. 1: MC57G murine fibroblasts were co-cultured for 48 hours with *T. cruzi* Tulahuen trypomastigotes. 2: Infected fibroblasts were harvested and lysed using a non-denaturing lysis buffer. 3: Immuno-affinity purification column was prepared by covalently linking MHC-I–specific mAbs (M1/42) to AminoLink Plus resin. 4: Lysed fibroblast material was loaded on the column, allowing the peptide-loaded MHC-I complexes to bind. 5: Four different wash buffers were used to remove the non-specific components. 6: Acetic acid was used to elute the peptide—MHC-I complexes from the column and dissociate the peptide from the MHC-I. 7: A spin filter column separated the peptide fraction from the MHC-I molecules. 8: Peptides were concentrated using a speedvac concentrator. 9: concentrated peptides were analyzed using LC-MS/MS for their sequence. 10: using the *T. cruzi* proteome from UniProt *T. cruzi* proteins were identified. Mouse self-proteins were identified using the C57BL/6J

proteome. Figure prepared with Biorender.com. B) *T. cruzi* infected and non-infected MC57G fibroblasts, as well as green monkey kidney (VERO) cells were stained with anti-mouse MHC-I Alexa Fluor 488. Cells were analyzed by flow cytometry and data was reported as median fluorescent intensity (MFI). Mean and standard deviation are shown and were calculated from six technical replicates. Statistical significance: **: $p < 0.01$. C) Number of MHC-I binding peptides identified from *T. cruzi* or mouse for each experiment. D) Distribution of the peptide length for each experiment.

9 aa (68.7%). This suggests an increase in the length of peptides presented on MHC-I during *T. cruzi* infection, a phenomenon that has been described previously for *T. gondii* infection [22].

## Peptides from Tcj2 (*T. cruzi* DnaJ 2) protein are presented on MHC-I from *T. cruzi*—Infected cells

24 unique *T. cruzi* peptides, originating from 17 unique proteins were identified from the two immunopeptidomics experiments with *T. cruzi* infected cells (Fig 3, more details in S1 Table). Notably, six different peptides were linked to *T. cruzi* DnaJ 2 (Tcj2) (Fig 3, Protein Group 2), with their peptide sequences located throughout the Tcj2 protein (S2 Fig). Spectral matching using synthetic peptides confirmed the identification of the six Tcj2 derived peptides (S3 Fig). Furthermore, the Tcj2 derived peptides AFYTGKTIKLA and VKETKFYDSLG were identified in both immunopeptidomics experiments. Tcj2 was the only *T. cruzi* protein that was identified by both experiments, increasing the confidence of being a protein that is well processed and presented on MHC-I of *T. cruzi* infected cells.

The DNAJ heat shock protein family, also known as heat shock protein 40 (HSP40), represents a highly conserved group of proteins throughout evolutionary history. Five different

| Protein Group | Peptide | Length | -10lgP | Unique | Identified in experiment # | Protein Accession | Protein Description |
|---|---|---|---|---|---|---|---|
| 1 | ASYDALETANKMGLL | 15 | 41.91 | Y | 1 | Q4E2Y1 | 60S ribosomal protein L23a  putative |
| 2 | LNIKHLDDRDVS | 12 | 34.91 | Y | 2 | Q4D832 | Heat shock protein DnaJ, putative |
| | VKETKFYDSLG | 11 | 33.21 | Y | 1+2 | | |
| | AFYTGKTIKLA | 11 | 32.99 | Y | 1+2 | | |
| | LEAFYTGKTIKLA | 13 | 31.76 | Y | 1 | | |
| | SNEISDLR | 8 | 22.18 | Y | 2 | | |
| | GEGDQIPGVR | 10 | 15.31 | Y | 2 | | |
| 3 | AEFAKKMEEQNKKFF | 15 | 32.11 | Y | 1 | Q4D3A5, Q4D3A7, Q4D7Y4 | Kinetoplastid membrane protein KMP-11 |
| 4 | MTYKPVIHGRPGVG | 14 | 31.24 | Y | 2 | Q4DGZ5 | 40S ribosomal protein S15, putative |
| | KERTFHKFTYRGLE | 14 | 18.43 | Y | 2 | | |
| 5 | LYALYRQKKEKPRN | 14 | 30.62 | Y | 2 | Q4DTQ1 | 40S ribosomal protein S23, putative |
| 6 | LFSGMKVLRLR | 11 | 30.3 | Y | 2 | Q4DY30, Q4DY32 | RNA-binding protein, putative |
| | LFSGMKVLR | 9 | 21.87 | Y | 2 | | |
| 7 | DVFVNGKKPVYD | 12 | 26.87 | Y | 2 | Q4DAW5 | Cytochrome C oxidase subunit VI, putative |
| 8 | IVPVPFIKV | 9 | 26.35 | Y | 2 | Q4D289 | Neurobeachin/beige-like protein, putative |
| 9 | SAAGATTLVENF | 12 | 24.47 | Y | 1 | Q4DRE3, Q4CW14 | Uncharacterized protein |
| 10 | LVRHMASKDRSARL | 14 | 23.6 | Y | 2 | Q4DCN9 | Glyceraldehyde-3-phosphate dehydrogenase |
| 11 | DQGSADIVN | 9 | 23.54 | Y | 2 | Q4D976, Q4E5Z1 | ATP-dependent DEAD/H RNA helicase, putative |
| 12 | AEFAESKV | 8 | 23.25 | Y | 2 | Q4DNZ9 | Nfu_N domain-containing protein |
| 13 | PAVVAPPPQ | 9 | 22.63 | Y | 2 | Q4CNA5, Q4DPR2 | OTU domain-containing protein |
| 14 | VAAKRSATSAKLG | 13 | 21.63 | Y | 2 | Q4CY60, Q4D3G1 | Uncharacterized protein |
| 15 | SSSEKDYYKILG | 12 | 20.75 | Y | 2 | Q4D7B1, Q4DVP6 | Chaperone DnaJ protein, putative |
| 16 | GVLATGASLA | 10 | 20.67 | Y | 2 | Q4DH33, Q4DY72 | Uncharacterized protein |
| 17 | ASVVAGNIS | 9 | 20.62 | Y | 2 | Q4DBX3 | Uncharacterized protein |

**Fig 3. List of all *T. cruzi* peptides presented on MHC-I and their source proteins.** The 24 unique peptides traced back to 17 *T. cruzi* proteins. The identified heat shock protein DnaJ (protein group 2) is one of 5 DnaJ (heat shock protein 40) proteins described for *T. cruzi* that is identified in literature as Tcj2.

DnaJ proteins have been described for *T. cruzi*, and the DnaJ-derived peptides found in our immunopeptidomics experiments, are derived from one of these DnaJ proteins, described as *T. cruzi* DnaJ 2 (Tcj2) [40,41]. In comparison, more than 40 DnaJ proteins have been described in humans [42]. They are molecular chaperones of heat shock protein 70 (HSP70) and are involved in protein (re)folding [40]. It has been described that DnaJ proteins are located in all cellular compartments of *T. cruzi*, and it has been suggested that DnaJs are potentially secreted proteins [43]. Furthermore, comparative proteome analysis of trypomastigote and amastigote stages of *T. cruzi* suggested that Tcj2 are similarly expressed in both stages of the parasite [44]. Notably, our immunopeptidomics data revealed the presence of a DnaJ chaperone protein (Fig 3, Protein Group 15). Unlike TcJ2, this chaperone protein, while related, is not classified as a DnaJ protein and exhibits a low sequence identity of 32.6% with Tcj2. In the context of vaccine development two critical aspects of a vaccine target come into play: its degree of conservation across diverse parasitic strains and its dissimilarity from host self-proteins, ensuring it does not provoke autoimmunity. For *T. cruzi* over 6000 cruzi strains, classified into six Discrete Typing Units (DTUs) have been described [45]. Therefore, Tcj2 protein sequence alignments were conducted to address these questions. When comparing the Tcj2 sequence between the different genetic groups of *T. cruzi*, it was observed that Tcj2 is very conserved (S1 Fig). Between 9 different *T. cruzi* strains from the three available DTUs, only twice a difference is a single residue was observed. Next, the amino acid sequence of Tcj2 was blasted using BLASTp and compared to human and mouse DnaJ proteins with the highest sequence identity to Tcj2 (S2 Fig). It can be observed that there is a considerable degree of identity (43.0% identity and 60.0% similarity) between human DnaJA4 and *T. cruzi* Tcj2 protein. However, enough differences in amino acids in the sequence are present which make overlapping T cell epitopes of 8–15 amino acids unlikely. DnaJ-homolog protein sequence between human and mouse showed a high degree of similarity (93.2% identity) which consequently, resulted in comparable similarities between mouse and *T. cruzi* (42.5% identity and 60.7% similarity) as observed for human and *T. cruzi*. Overall, this suggests a reduced risk of auto-immune reaction induction to host DnaJ through complete amino acid sequence identity when using *T. cruzi* Tcj2 (DnaJ) as a vaccine target [46]. Additionally, sequence alignments between Tcj2 (DnaJ) and DnaJ-homologs from other trypanosomes were performed. Sequence identity showed a considerable degree of overlap for *T. brucei brucei* (72.41% identity) (S4 Fig), as well as for *T. congolense* (71.79% identity) and *T. vivax* (72.10% identity). *T. rangeli* showed a lower degree of overlap with *T. cruzi* with 45.69% identity. For *T. brucei brucei*, *T. congolense* and *T. vivax*, multiple stretches of identical amino acid sequences are present which could serve as overlapping epitopes for antibodies and CTLs. For example, the peptide sequence GEGDQIPGVR from *T. cruzi* Tcj2 found by immunopeptidomics is also present in the DnaJ-homolog of *T. brucei brucei* and *T. vivax*. Overall, these observations suggest that a *T. cruzi* Tcj2 (DnaJ) vaccine candidate antigen has the potential to cross-protect against other trypanosomes infections.

### *In vitro* immunogenicity testing of Tcj2-expressing mRNA vaccine

An mRNA vaccine construct was designed based on the Tcj2 protein sequence from *T. cruzi* (Fig 4A). On the 3'–end of the Tcj2 sequence, the amino acid sequence SIINFEKL was added, followed by a FLAG-tag. SIINFEKL (OVA 257–264) is a sequence naturally occurring in the chicken ovalbumin protein that has a high affinity to H-2K$^b$ (MHC-I expressed by C57BL/6J mice). An antibody is available that allows detection of MHC-I antigen presentation of this model epitope, as well as MHC-I/SIINFEKL tetramers for the quantification of SIINFEKL-specific CD8+ T cells [47]. A FLAG-tag was added to evaluate the translation of the complete

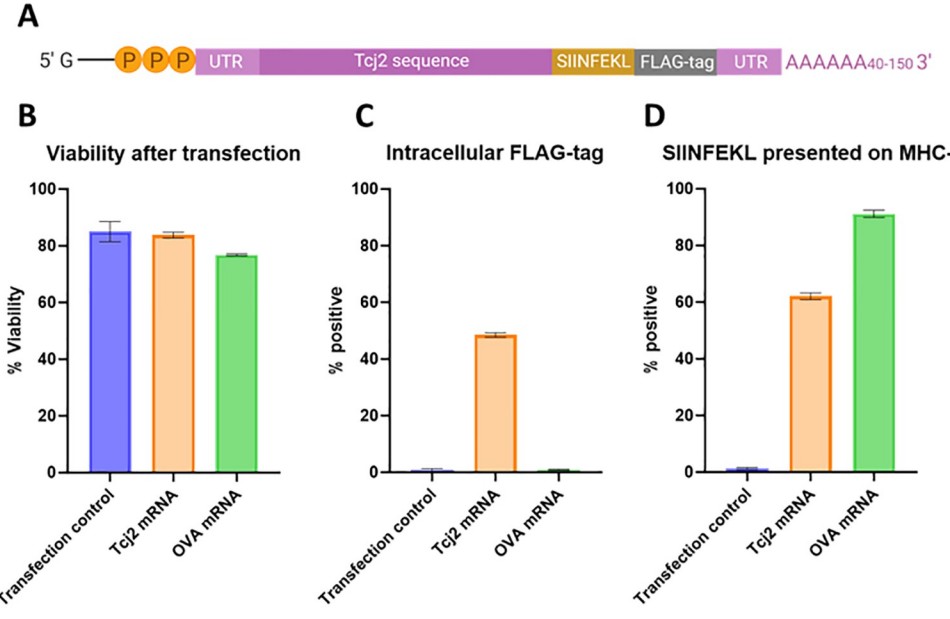

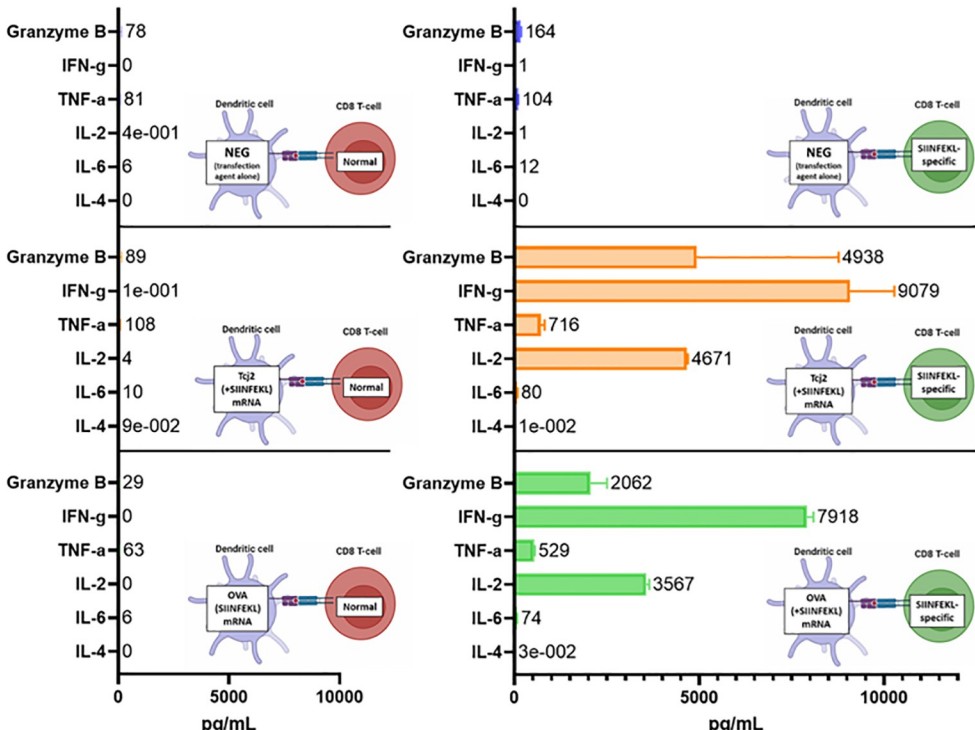

**Fig 4.** *In vitro* **evaluation of Tcj2 mRNA showed the translation and antigen presentation of the mRNA construct, as well as the activation of SIINFEKL-specific CD8+ T cells.** DC2.4 cells were transfected with mRNA with Lipofectamine MessengerMAX, or without mRNA (transfection control). After a 24-hour incubation, cells were subjected to analysis. A) Schematic representation of Tcj2 mRNA construct. B) Cell viability measured after transfection. C) Detection of the translated FLAG-tag sequence by intracellular flow cytometry staining using a FLAG-specific antibody. D) Presentation of SIINFEKL on surface MHC-I (H-2K[b]) measured by flow cytometry using an antibody specific for the combination of SIINFEKL presented by H-2K[b]. E) Cytokines secreted by C57BL/6J "normal" or SIINFEKL specific OT-1 CD8+ T cells after co-culture with transfected cells. DC2.4 cells were transfected for 24 hours and then counted and seeded. Splenocytes were added in a ratio of 1:10 (DC2.4: splenocytes) and the co-culture was incubated for 24 hours before the supernatant was collected and analyzed for cytokines by Luminex.

Figure prepared with Biorender.com. From all experiments, mean and standard deviations are shown from triplicate experiments.

mRNA construct, which can be measured intracellularly by flow cytometry using a FLAG-tag specific antibody.

After a 24-hour transfection with Tcj2 mRNA and Lipofectamine MessengerMAX, the viability of the mouse dendritic cell line DC2.4 cells was analyzed. Results (Fig 4B) showed that Tcj2 mRNA has no impact on the viability of the cells compared with transfection agent only (84% versus 85%, respectively). Transfection with Ovalbumin mRNA, which was used as a positive control for the MHC-I presentation of SIINFEKL peptide, decreased the viability slightly to 77%. More than 40% of the Tcj2 mRNA transfected cells were positive for intracellular FLAG-tag staining of this epitope located at the C-terminal end of the protein. This shows that the Tcj2 mRNA was completely translated (Fig 4C). As expected, no FLAG-tag was detected in cells transfected with Ovalbumin mRNA or in transfection control cells. Additionally, almost 60% of the cells transfected with the Tcj2 mRNA were positive for SIINFEKL presentation on MHC-I, as revealed by the H-2K$^b$ SIINFEKL specific antibody (Fig 4D), followed by 80% for the positive Ovalbumin mRNA control. These results show that the Tcj2 mRNA is translated and that peptides derived from the produced protein are presented on MHC-I. Furthermore, when Tcj2 mRNA transfected DC2.4 cells were co-cultured with splenocytes from transgenic C57BL/6J OT-1 mice, which contain high numbers of CD8+ T cells specific for the SIINFEKL epitope presented on H-2K$^b$, an increase in Granzyme B, IFN-γ, TNF-α and IL-2 was measured (Fig 4E). Production of IL-2 suggests the activation of T cells and Granzyme B, IFN-γ and TNF-α hallmarks the activation of CD8+ cytotoxic T cells. Moreover, the results looked similar in the co-culture where DC.24 cells were transfected with Ovalbumin mRNA, demonstrating that the SIINFEKL peptide presented on MHC-I is responsible for activating the OT-1 CD8+ T cells. No increase in Granzyme B, IFN-γ, TNF-α and IL-2 was measured when splenocytes from non-transgenic (= "normal") C57BL/6J mice were used. Overall, the data suggests that the Tcj2 mRNA construct was translated *in vitro*, antigen presented on MHC-I, and induced CD8+ T cell activation of naïve antigen specific T cells.

## Preparation of Tcj2 mRNA LNPs

Following *in vitro* validation of the translatability of the Tcj2-encoding mRNA construct and its subsequent presentation on MHC-I and activation of antigen-specific CD8+ T cells, Tcj2-encoding mRNA was formulated in lipid nanoparticles (LNPs) for further *in vivo* assessment of this mRNA vaccine. To serve as a negative control, LNPs without mRNA (empty LNPs) were also prepared. Based on the Ribogreen assay, the loading efficiency of the encapsulated mRNA was 85%. Dynamic Light Scattering (DLS) analysis indicated that both Tcj2 mRNA LNPs and empty LNPs exhibited sizes within the range of 80–90 nm, which is considered suitable for mRNA LNP immunogenicity studies (Fig 5A) [48]. Freezing at -80˚C followed by thawing did not significantly affect the LNP diameter (Z-ave), with measurements of 83.7 d.nm versus 84.7 d.nm for Tcj2 mRNA LNPs and 82.9 d.nm versus 87.9 d.nm for empty LNPs before and after freezing, respectively. Furthermore, polydispersity index (PdI) of Tcj2 mRNA LNPs was measured to be low before and after freeze/thaw, indicating a small variation in particle size (Fig 5A and 5B). For the empty LNPs a PdI of 0.331 was measured, surpassing the acceptance criterion of 0.25, although it decreased to 0.189 after freeze/thaw. The higher PdI values observed in the empty LNPs can be attributed to the absence of mRNA in the formulation, which is crucial for proper LNP formation. To further characterize the Tcj2 mRNA

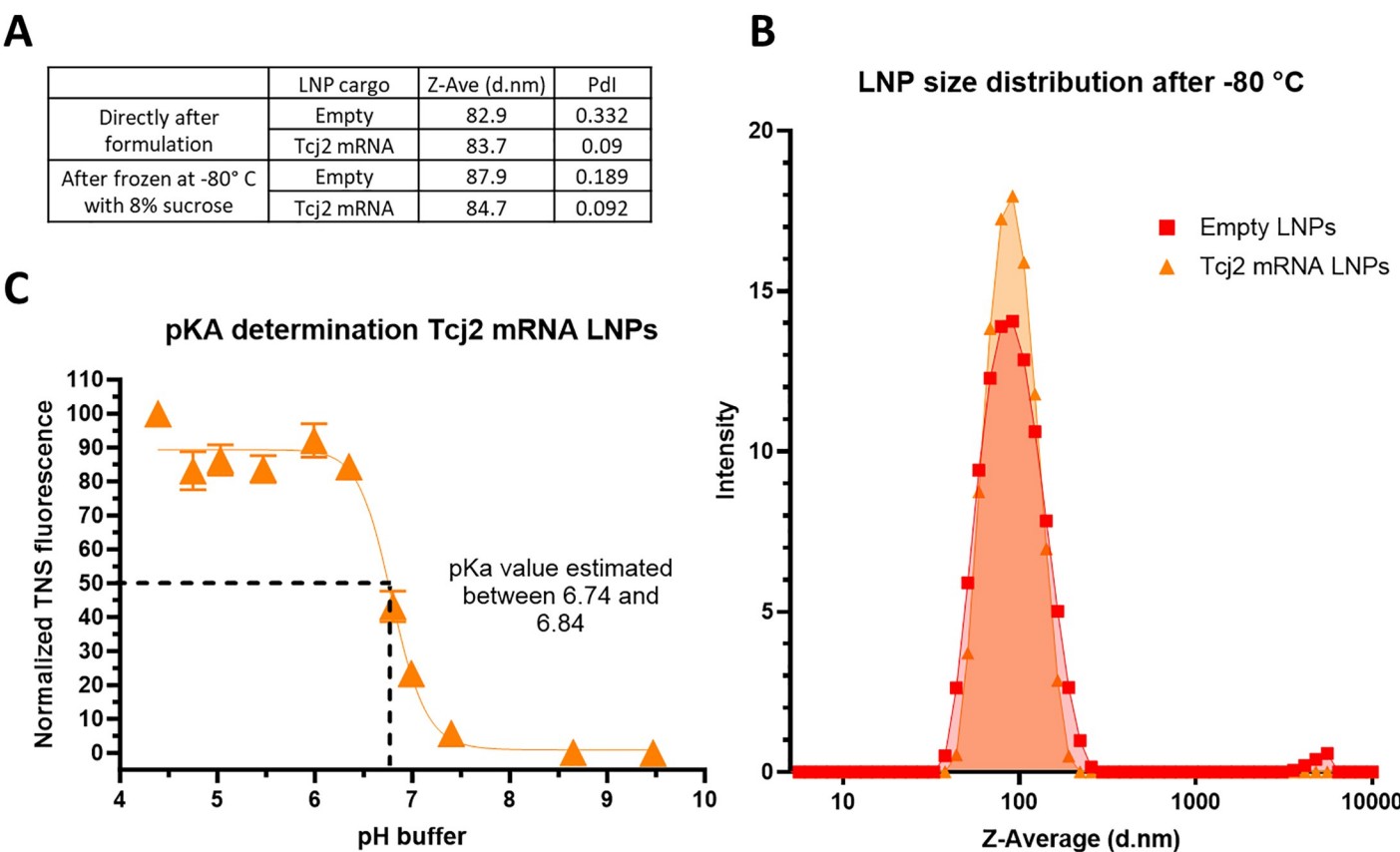

**Fig 5. Freeze/thawed Tcj2 mRNA LNPs showed desired diameter, polydispersity index and pKa range for immunogenicity studies.** A) Tcj2 mRNA LNPs and empty LNPs were analyzed by Dynamic Light Scattering (DLS), directly after formulation and after storage at -80° C. Diameter in nm (Z-Ave (d.nm)) and polydispersity index (PdI) of LNPs were reported. B) Distribution of the size of the Tcj2 mRNA LNPs and empty LNPs. C) pKa of the surface of Tcj2 mRNA LNPs was analysed though a TNS fluorescence assay.

LNPs and determine whether they are suitable for intramuscular (IM) delivery, the surface charge of the LNPs was examined by measuring the pKa (acid dissociation constant). The pKa of LNPs has been previously shown to be a determining factor for immunogenicity, influencing factors such as endosomal escape delivery and cellular uptake [49]. As shown in Fig 5C, the pKa of the LNP's surface falls within a range of 6.740 and 6.841, what is within the optimal pKa range of 6.6–6.9 for IM delivery [49]. In summary, results indicated that the mRNA encapsulation, the diameter, the polydispersity index, and the surface charge of the LNPs all fall within the desired range for LNP mRNA vaccines.

## Tcj2 mRNA LNPs elicit robust antigen-specific T cells and humoral immune responses

To evaluate the immunogenicity of the Tcj2 mRNA vaccine candidate, five mice were immunized with Tcj2 mRNA LNPs on day 0 followed by a boost at day 21 (Fig 6A). As a control, five mice were immunized with empty LNPs. At day 40, mice were euthanized, and sera and spleens were collected for immune evaluation. The *in vivo* study was conducted twice to show reproducibility.

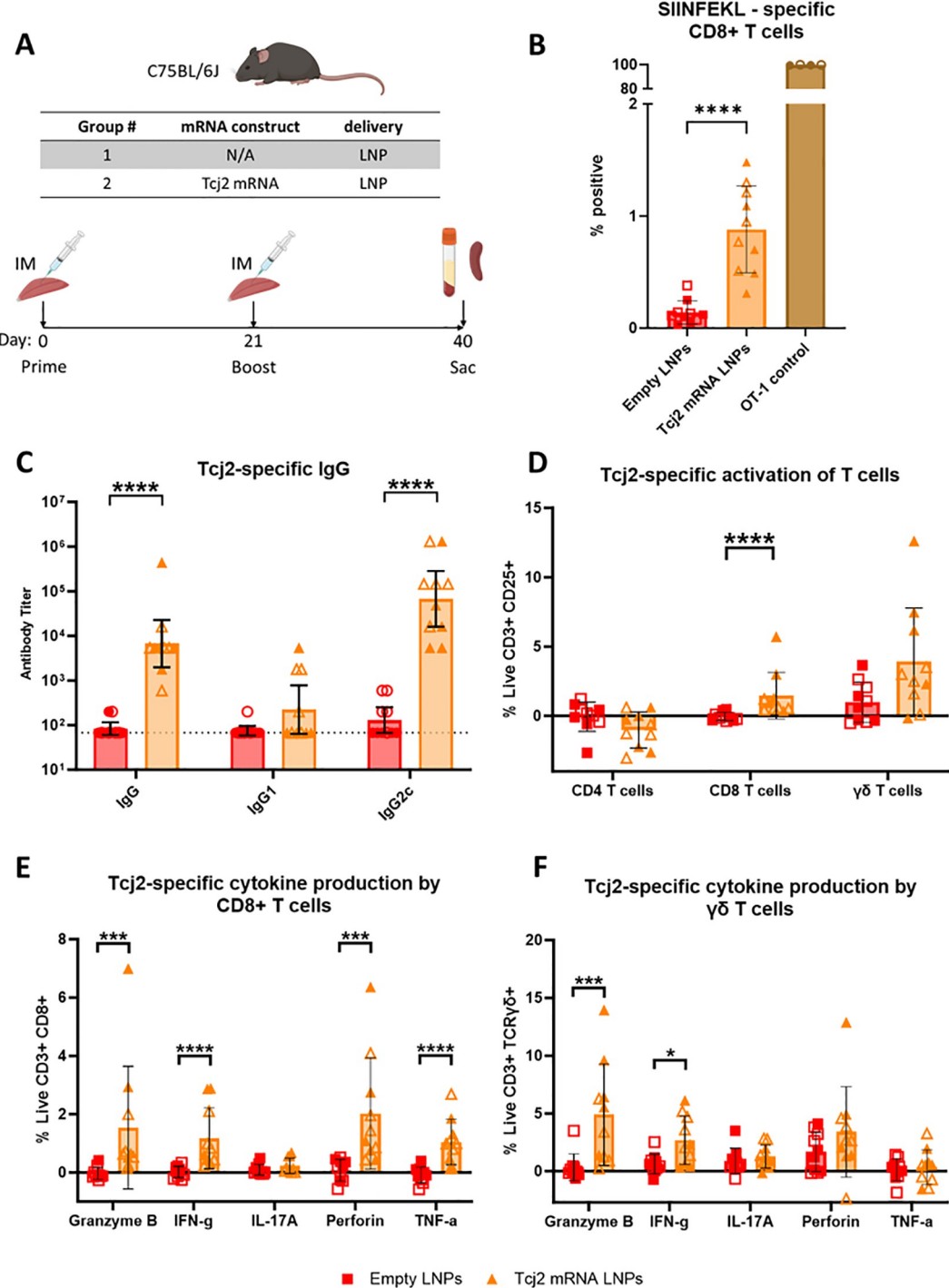

**Fig 6. Tcj2 mRNA LNPs elicited humoral and cellular immune responses in a mouse immunogenicity study.** A) Schematic description of mouse model, study groups and immunization schedule. Figure prepared with Biorender.com. B) SIINFEKL-specific CD8+ T cells measured by SIINFEKL tetramer staining. C) Tcj2-specific IgG, IgG1 and IgG2c was measured using an indirect ELISA coated with rTcj2 protein. The baseline antibody titer was set at 67 and is highlighted by the dotted line. D) Antigen-specific activation of T cells was measured after *in vitro* restimulation with rTcj2 protein by analyzing CD25 late activation marker expression. E) Cytokines and cytolytic compounds produced by CD8+ T cells after *in vitro* restimulation with rTcj2 protein. F) Cytokines produced by γδ T cells after *in vitro* restimulation with rTcj2 protein. For data presented in D, E and F: values from non-stimulated cells were subtracted from rTcj2 protein stimulated cells to obtain antigen-specific cytokine production. Statistical significance: *: $p < 0.05$, **: $p < 0.01$, ***: $p < 0.001$. For panel B, D, E and F

the mean and standard deviation are shown, while for C the geometric mean with 95% confidence intervals in shown. Filled symbol shapes represent *in vivo* study #1, while open symbol shapes represent repeat *in vivo* study #2.

To evaluate whether the Tcj2 mRNA had elicited the generation of antigen-specific T cells, the percentage of SIINFEKL specific CD8+ T cells in splenocytes of immunized mice (Fig 6B) was determined by SIINFEKL tetramer staining. Immunizations with Tcj2 mRNA LNPs resulted in a significant increase in SIINFEKL-specific CD8+ T cells compared to empty LNPs (mean 0.88% for Tcj2 mRNA LNPs versus 0.14% for empty LNPs). The location of the mRNA sequence encoding for SIINFEKL at the 3' end of the mRNA construct indicates successful delivery of Tcj2 mRNA to antigen-presenting cells (APCs), subsequent translation into protein, and processing for presentation on MHC-I to CD8+ T cells, leading to the generation of SIINFEKL-specific CD8+ T cells and potentially Tcj2-specific CD8+ T cells. Furthermore, splenocytes from the transgenic OT-1 mice, used as a positive control for SIINFEKL tetramer staining, exhibited nearly all CD8+ T cells being SIINFEKL specific (mean 99.7%).

To assess Tcj2 antigen-specific responses, recombinant Tcj2 (rTcj2) was prepared in *E. coli* (S5 Fig). The protein was then utilized in ELISAs to determine the generation of Tcj2-specific antibodies and in *in vitro* restimulation assays to evaluate antigen-specific T cell responses. Immunizations with Tcj2 mRNA LNPs induced a significant increase in Tcj2-specific IgG and IgG2c titers, while specific IgG1 titers were not significantly increased compared to the empty LNP control (Fig 6C). The strong increase in IgG2c but not IgG1 suggest a skewed humoral immune response towards Th1 [50]. Testing *T. cruzi* trypomastigote lysate on western blot with sera from Tcj2 mRNA LNP immunized mice confirmed that the induced Tcj2 antibodies recognize native Tcj2 from *T. cruzi* (S6 Fig). Importantly, native DnaJ in lysates from uninfected mouse and human cell lines was not recognized, indicating that the Tcj2-specific antibodies do not cross-react with human and mouse DnaJ. Presence of human DNAJA4, which of all human DnaJ's shares the highest protein sequence identity with Tcj2, was confirmed in the HEK293T lysate by western blot (S7 Fig). Furthermore, Tcj2 antisera did also not show cross-reactivity with the much more homologous DnaJ in *T. brucei* lysate (S6 Fig).

To assess the cellular immune response against Tcj2, splenocytes were *in vitro* restimulated for 48 hours with rTcj2 protein, followed by analysis by flow cytometry. The expression of CD25, a late activation marker expressed on T cells, was significantly upregulated by CD8+ T cells after restimulation, as well as an observable trend in increase for γδ T cells (Fig 6D). CD4 + T cells displayed minimal changes in CD25 expression, implying a lesser role in the immune response against Tcj2 when compared to CD8+ T cells and γδ T cells. This was further observed when the production of intracellular cytokines IFN-γ, TNF-α, IL-17A, and cytolytic compounds granzyme B and perforin were analyzed. Significant increases in production of granzyme B, IFN-γ, perforin and TNF-α were observed in CD8+ T cells (Fig 6E). For γδ T cells, a significant increase in the production of IFN-γ and Granzyme B was observed (Fig 6F). No significant changes in cytokine production were observed by CD4+ T cells after restimulation (S8 Fig) but an observable trend in increase in IFN-γ was observed. These findings suggest that CD8+ T cells and γδ T cells were stronger activated by Tcj2 mRNA LNP compared to CD4+ T cells.

## Immunizations with Tcj2 mRNA LNPs induced significant memory CD8+ T cells response with cytotoxic features

To improve our understanding of the antigen-specific CD8+ T cell response induced by immunization with Tcj2 mRNA LNPs, central and memory CD8+ T cells were examined for

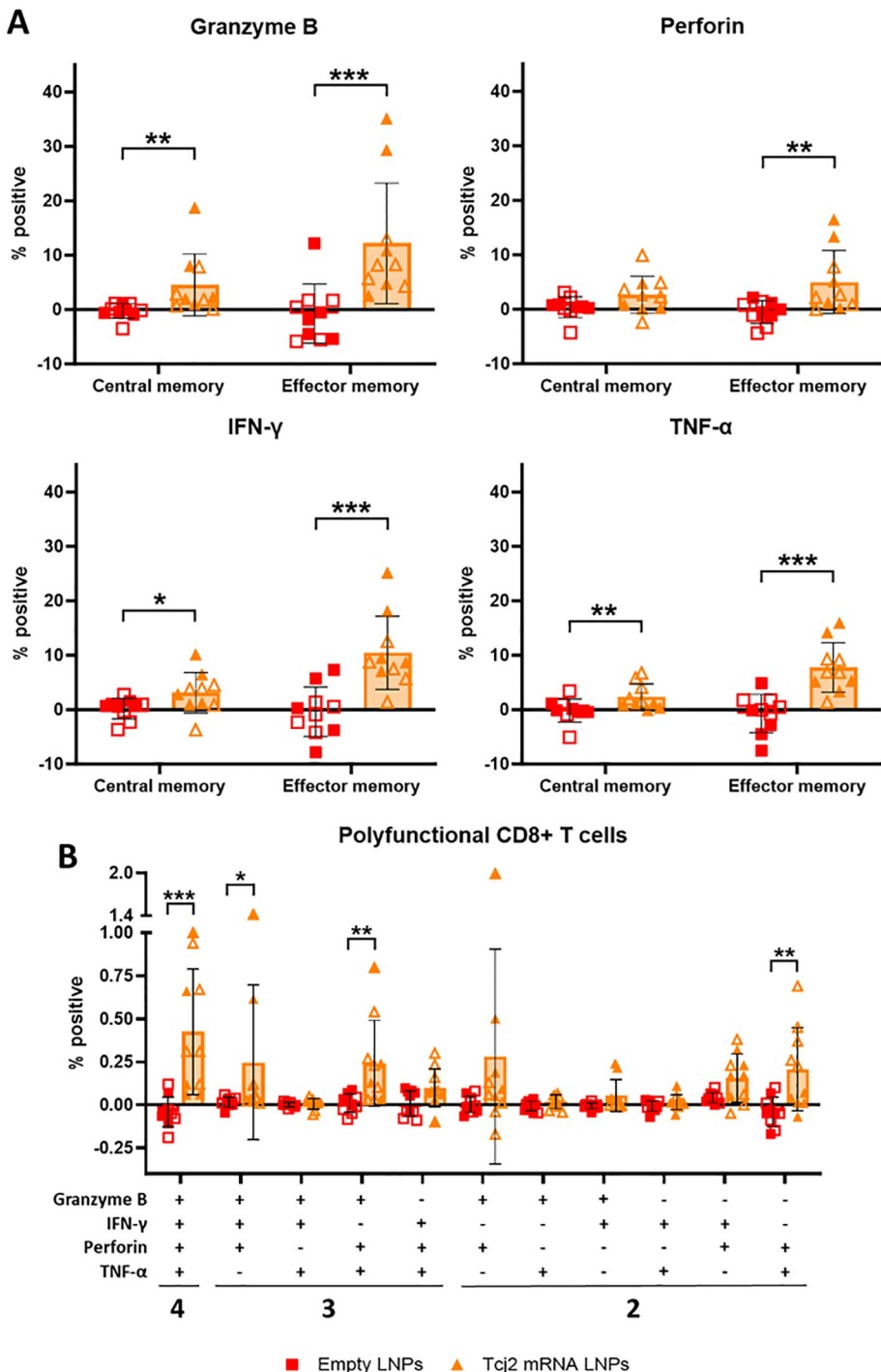

**Fig 7. Tcj2 mRNA LNP immunizations induced antigen-specific cytokine production in central memory and effector memory CD8+ T cells, as well as increase in cytokine producing polyfunctional CD8+ T cells.** A) antigen-specific central memory (CD62high and CD44high) and effector memory (CD62low and CD44high) CD8+ T cells were analyzed for production of granzyme B, perforin, IFN-γ or TNF-α. B) Antigen-specific CD8+ T cells were analyzed for the production of two or more cytolytic enzymes or cytokines using a Boolean combination gate strategy. For data presented in A and B: values from non-stimulated cells were subtracted from rTcj2 protein stimulated cells to obtain antigen-specific cytokine production. Statistical significance: *: p < 0.05, **: p < 0.01. For all panels in this figure the mean and standard deviation are shown. Filled symbol shapes represent *in vivo* study #1, while open symbol shapes represent repeat *in vivo* study #2.

their production of cytolytic enzymes and cytokines after *in vitro* restimulation with rTcj2. Both central and effector memory CD8+ T cells producing granzyme B, IFN-γ or TNF-α were significantly increased (Fig 7A), and effector memory CD8+ T cells producing perforin were also increased significantly. Additionally, polyfunctionality of CD8+ T cells was investigated, as measured by the ability to produce more than one cytolytic enzyme or cytokine (Fig 7B). The results showed a statistically significant increase of polyfunctional CD8+ T cells producing granzyme B, perforin, IFN-γ and TNF-α compared to the empty LNP control. Also, CD8+ T cells producing granzyme B, perforin and IFN-γ were significantly increased, as well as cells producing granzyme B, perforin and TNF-α. Furthermore, CD8+ T cells producing perforin and TNF-α simultaneously were also significantly increased. To summarize, immunizations with Tcj2 mRNA LNPs led to a robust increase in central and effector memory CD8+ T cells that produce cytolytic enzymes and cytokines upon restimulation, as well as an increase in CD8+ T cells that demonstrate polyfunctionality.

## Splenocytes from Tcj2 mRNA LNPs immunized mice decreased *T. cruzi* parasite load *in vitro*

To evaluate whether the observed cytotoxic properties from the CD8+ T cells would show functional protectiveness against *T. cruzi* infection, splenocytes from Tcj2 mRNA LNP–immunized mice were co-cultured with *T. cruzi*–infected MC75G mouse fibroblasts. A transgenic *T. cruzi* (Tulahuen, clone C4 +*lac*Z) parasite strain was used that expresses β-galactosidase, which allows for the utilization of a colorimetric reaction with chlorophenol red-β-D-galactopyranoside [51]. The enzymatic activity is directly proportional to the number of parasites, and can therefore be used to measure parasite load *in vitro*. The results of the co-culture between *T. cruzi*–infected fibroblasts and splenocytes showed a significant reduction of 13.7% in β-galactosidase activity compared to splenocytes from empty LNP-immunized mice (3.1%), where 100% reduction was achieved after *in vitro* treatment with 100 μM benznidazole (Fig 8). This observation suggests that parasite load was reduced by Tcj2 mRNA LNP-immunized splenocytes.

## Discussion

Mass-Spectrometry based immunopeptidomics has emerged as a powerful tool to identify new vaccine targets for cancers and infectious pathogens. For Chagas disease, a parasitic infection characterized by intracellular replication of *T. cruzi*, it is most relevant to analyze the peptides presented on MHC-I, since these peptides can be recognized by CD8+ T cells and are a target for CTLs to eliminate infected cells [11,52]. Most proteins from *T. cruzi* are unsuitable for vaccine development in stimulating a defensive cellular immune response, as they are unlikely to efficiently engage the MHC-I antigen presentation pathway upon infection. Utilizing immunopeptidomics data, we can pinpoint the proteins that are capable of MHC-I presentation and subsequent recognition by CTLs.

Analysis of the immunopeptidome of *T. cruzi* infected fibroblasts revealed 24 unique *T. cruzi* peptides presented on MHC-I, along with over one thousand murine self-peptides. H2-K^b and H2-D^b MHC-I haplotypes were immunoaffinity isolated from *T. cruzi* infected and uninfected murine MC57G fibroblast cells, using the anti-H2 M1/42 mAb [53]. Upon *T. cruzi* infection of fibroblasts, an increase in M1/42 binding to the cells was observed, demonstrating an upregulation of MHC-I presentation. Contradicting reports suggesting MHC-I downregulation by *T. cruzi* as a strategy to evade T cell immune detection, while upregulation has also been observed [54–56]. The upregulation of MHC-I molecules observed in our study may also

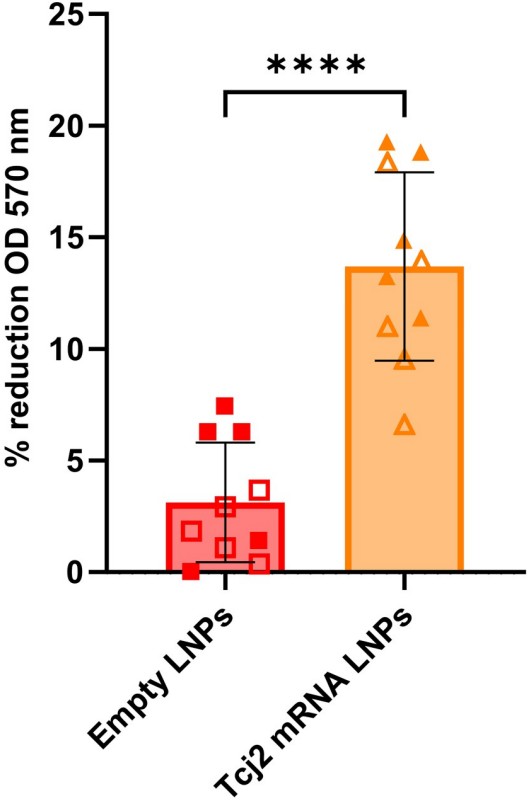

**Fig 8. Splenocytes from Tcj2 mRNA LNP—Immunized mice decrease *T. cruzi* infection *in vitro*.** *T. cruzi*–infected MC57G mouse fibroblasts were co-cultured with splenocytes from immunized mice for 72 hours. *T. cruzi* parasites expressed the LacZ gene which encodes for the enzyme β-galactosidase, used to convert a substrate which directly correlated to the parasite load in each well. *T. cruzi*–infected fibroblasts treated with a high dose of benznidazole were run in parallel and used to calculate the %-reduction of parasite load. Statistical significance: **: $p < 0.0001$. Mean and standard deviation are shown. Filled symbol shapes represent *in vivo* study #1, while open symbol shapes represent repeat *in vivo* study #2.

explain the higher abundance of mouse self-peptides in the infected fibroblast samples compared to the uninfected sample.

The identified 24 unique MHC-I presented *T. cruzi* peptides were originating from 17 distinct proteins, reflecting the availability of these proteins for processing and presentation on MHC-I of murine fibroblasts. Notably, Tcj2 protein was identified as the source of six different peptides, and Tcj2 was the only protein identified in both immunopeptidomics experiments, making Tcj2 our most promising vaccine candidate antigen for inducing MHC-I restricted cytotoxic T cells. Of the other 17 proteins, several are known to be involved in intracellular endogenous processes, such as ribosomal proteins, oxidases, dehydrogenases, helicases, and heat shock proteins, indicating their intracellular localization and functions. However, KMP-11, one of the 17 identified proteins, is a membrane protein that has been shown to be at least partially exposed on the surface of *T. cruzi* trypomastigotes, since KMP-11 antisera can affect

parasite invasion and flagellar motility [57]. Three proteins remain uncharacterized even after blasting their protein sequence, leaving their subcellular localization and functions to be determined. Remarkably, no trans-sialidases derived peptides were found, while it has been observed that these proteins are highly expressed in trypomastigotes, located on the outside of the plasma membrane, sometimes GPI-anchored, and are immunodominant for CD8+ T cells during natural infection [58,59]. Given that trans-sialidase–specific CD8+ T cells did not recognize *T. cruzi*–infected cells within the first 48 hours after infection in previous research, the timing of our immunopeptidomics MHC-I profiling of *T. cruzi* at 48 post infection may have been too closely aligned with this timeframe, potentially contributing to the absence or limited availability of trans-sialidase peptides [60].

We further focused on the potential of Tcj2 as a promising vaccine candidate. We assessed the antigen's conservation across various Trypanosomes and explored potential cross-reactivity with heat shock proteins in mice and humans. Sequence alignments revealed minimal overlapping peptide sequences shared by Tcj2 and mouse or human DnaJ, suggesting a reduced likelihood of inducing autoimmune reactions to self-DnaJ by Tcj2 mRNA vaccination in mice and humans. Furthermore, during the full duration of the *in vivo* immunogenicity study, the Tcj2 mRNA LNPs injected mice appeared clinically normal, supporting the safety profile of the Tcj2 mRNA vaccine. Additionally, Tcj2-specific antibodies generated through immunizations did not cross-react with native DnaJ expressed in mouse MC57G or human HEK293 cell lines, nor with native DnaJ from *Trypanosoma brucei* [61–63]. Protein sequence alignments showed the existence of multiple stretches of identical amino acid sequences between DnaJ from *T. cruzi*, *T. brucei brucei*, *T. congolense* and *T. vivax*, suggesting there are potential overlapping epitopes for CTL for cross-protection between these different trypanosomes. Considering the conserved nature of Tcj2 across different *T. cruzi* DTUs, protection against all *T. cruzi* strains could be possible, but this will be dependent on the expression profile of Tcj2 between different *T. cruzi* DTUs.

Heat shock proteins are critical for maintaining the structural integrity of cellular protein machinery and function in protein folding, as well as in the degradation of misfolded proteins. In protozoans, such as *T. cruzi*, heat shock proteins like Tcj2 play vital roles in the parasite's ability to adapt to hostile and constantly changing environments. *T. cruzi* undergoes transitions between hosts and transmission vectors throughout its lifecycle, as well as switches between intracellular and extracellular stages. These environmental changes encompass variations in pH levels, temperature, oxidative stress, and immune responses [64]. Therefore, it comes to no surprise that Tcj2 is described in literature to be expressed in different compartments of the *T. cruzi* trypomastigotes and amastigotes [40,44].

Vaccines targeting DnaJ proteins like Tcj2 have previously demonstrated effectiveness against various microbial pathogens. In the context of a vaccine candidate against typhoid, Saghi et al. administered recombinant DnaJ from *Salmonella enterica serovar Typhi* to mice and observed that this immunization approach induced robust humoral and cellular immune responses, resulting in 70% protection against a lethal challenge with *Salmonella typhimurium*, highlighting the potential of DnaJ as a vaccine target [65]. Similarly, in the case of *Ureaplasma urealyticum* infection, DnaJ—immunizations have been shown to elicit a strong humoral and a Th1-mediated cellular immune response, along with a significant decrease in bacterial load and inflammation in the reproductive tract of mice [66]. Moreover, Khan et al. demonstrated that vaccination with recombinant DnaJ protein from *Streptococcus pneumoniae* provided protection to 70% of mice against a lethal intranasal challenge with *S. pneumoniae* [67]. These findings collectively underscore the potential effectiveness of DnaJ proteins as a vaccine target against diverse pathogens.

We used our mRNA platform to evaluate the potential of Tcj2 as a vaccine against Chagas disease. *In vitro* assessments confirmed successful translation, MHC-I antigen presentation, and activation of antigen specific CD8+ T cells. When injected in C57BL/6J mice, Tcj2 mRNA induced robust humoral and cellular immune responses, including significant increases in cytotoxic T lymphocytes (CTLs) and associated cytotoxic enzymes and cytokines. Additionally, an *in vitro* killing assay demonstrated that splenocytes from Tcj2 mRNA immunized mice reduced *T. cruzi* infection, indicating the vaccine-induced effector functions. mRNA vaccines have demonstrated remarkable success in eliciting CD8+ T cell responses by efficiently presenting peptides on MHC-I to CD8+ T cells. Given the critical role of these T cells in identifying and killing parasite-infected cells, the mRNA vaccine platform presents a promising avenue for the creation of a Chagas disease vaccine [30]. The Tcj2 mRNA construct was designed with a SIINFEKL-tag to assess MHC-I antigen presentation and CD8+T cell activation *in vitro*, along with a FLAG-tag to measure intracellular expression. These tags are very useful for development and evaluation of the mRNA vaccine constructs, but the case of clinical evaluations, these will be removed from the mRNA construct. Notably, high IgG2c antibody titers specific to the rTcj2 protein were detected, and these antibodies were also capable of binding native *T. cruzi* Tcj2 on western blot. Importantly, immune clearance of *T. cruzi* trypomastigotes has been largely attributed to the IgG2 isotype [68]. However, it remains unknown whether Tcj2 is exposed on the surface of trypomastigotes and if antibodies could opsonize parasites. Furthermore, Tcj2 has been found to be secreted by trypomastigotes from different strains (Colombiana—TcI, Y strain—TcII, and CL Brener—TcVI) as part of the trypomastigote-derived secretome [69]. These secreted proteins could potentially be captured by Tcj2-specific antibodies, and processed by the MHC-I pathway when parasites are intracellular.

Evaluation of the cellular responses following Tcj2 mRNA LNP immunizations revealed the strongest induction of CD8+ T cells, followed by γδ T cells, while CD4+ T cells exhibited the lowest response. Antigen-specific central and effector memory CD8+ T cells produced cytotoxic enzymes granzyme B and perforin, as well as cytokines IFN-γ and TNF-α. The expansion of central memory CD8+ T cells is important for long-term immunological protection [70], while effector CD8+ T cells play a critical role in controlling ongoing *T. cruzi* infection [71]. Moreover, there was also an increase in antigen-specific polyfunctional CD8+ T cells capable of secreting multiple cytokines, indicating a robust T cell response with potential immunological control against infectious diseases [72]. Particularly for *T. cruzi* infection, polyfunctional T cell responses have been linked to effective anti-parasitic cell mediated immunity [73,74].

In the case of γδ T cells, there was a significant increase in granzyme B-secreting cells upon *in vitro* restimulation with rTcj2. Although γδ T cells have been studied less extensively than CD4+ and CD8+ T cells due to their unique properties, they have been implicated in parasitic control during the acute phase of *T. cruzi* infection [75]. γδ T cells can act as modulators of Th1 responses and exert infection through cytotoxic action against infected cells [76]. Similar observations have been made in other protozoan infections, such as *Plasmodium falciparum*, where they demonstrated the ability to recognize parasite-infected cells and eliminate them through the release of granzyme B [77]. This suggests the potential of the increased granzyme B-producing γδ T cells to exhibit similar effector functions in the context of *T. cruzi* infection.

We observed a minimal increase in antigen—specific CD4+ T cells upon *in vitro* restimulation, suggesting a low level of CD4+ T cell activation in response to the vaccine. This indicates that antigen processing and presentation via MHC-II might be occurring to a lesser extent, although the presence of Tcj2-specific IgG antibodies suggest the help and presence

of Tcj2-specific CD4+ T helper cells. The Tcj2 mRNA vaccine is primarily expected to undergo cytosolic antigen processing and presentation on MHC-I, leading to the activation of CD8+ T cells [78]. However, the observable trend in increase of CD4+ T cells producing granzyme B and IFN-γ after restimulation with rTcj2, as well as the presence of Tcj2-specific IgG2c antibodies in the sera, suggests also processing and presentation of Tcj2 through the MHC-II pathway. The observed low level induction of antigen-specific CD4+ T cells could be attributed to factors such as the mRNA delivery, the vaccination route, and the immunization schedule, as similar findings have been reported by Peng et al., showing a 10-fold lower activation of CD4+ T cells compared to CD8+ T cells upon antigen-specific restimulation [79].

This study has some limitations. i) MHC-I immunopeptidome screening: The reported experiments were limited to the screening of peptides presented by mouse fibroblasts (a matching model for our *in vivo* experiments). Human cell lines can be considered for complementary studies as well as different cell types, such as skeletal and cardiac muscle cells. The authors acknowledge the possibility that additional replicate studies, potentially at different times of infection, could unveil more peptides and identify additional vaccine antigen candidates. ii) Cross-reactive epitopes: structural similarities between peptides from Tcj2 and DnaJ proteins from mouse/human have not been evaluated, hence the risk of cross-reactivity cannot be excluded without further investigation. iii) The Tcj2 mRNA vaccine immunogenicity study: Albeit the in vivo study was conducted twice to show reproducibility, the group sizes were limited, and only one single formulation and a single dose was tested. Further investigations are essential for demonstrating the effectiveness of the Tcj2 mRNA vaccine candidate, including *in vivo* challenge studies and detailed cytokine analysis.

Although constrained by the mentioned limitations, our data showcases a pioneering application of immunopeptidomics in identifying novel vaccine targets for Chagas disease. Moreover, it underscores the mRNA platform's potential for rapid evaluation of new candidate antigens that stimulate antigen-specific CD8+ T cell responses. Further expanding on this work will increase the understanding of MHC-I presentation of *T. cruzi* antigens and identify the dominant and subdominant protein peptides recognized by CTLs, which may result in more effective vaccines. Based on the *in vitro* and *in vivo* data presented in this study, we consider Tcj2 a potentially strong vaccine antigen for Chagas disease and are currently planning challenge experiments using *T. cruzi* of different DTUs.

## Materials and methods

### Ethics statement

Animal experiments were performed in full compliance with the Public Health Service Policy and the National Institutes of Health Guide for the Care and Use of Laboratory Animals, 8th edition, under a protocol approved by Baylor College of Medicine's Institutional Animal Care and Use Committee, assurance number D16-00475 [80].

### Animals used for studies

Female C57BL/6J and C57BL/6J OT-1 mice were obtained at 5–6 weeks of age from The Jackson Laboratory and allowed to acclimate for one week prior to any manipulation. Mice were housed in groups of 4 in small microisolator caging, with ad libitum food and water and a 12hr light/dark cycle.

## Immunopeptidomics

**Preparation of *T. cruzi* trypomastigotes.** To obtain parasites for the immunopeptidomics experiments, 500 cm$^2$ TripleFlask treated cell culture flasks (ThermoFisher Scientific, Cat# 132913) were seeded with 5 x 10$^6$ VERO cells (ATCC, Cat# CCL-81) in cMEM media (MEM + 2% FBS + 1x pen/strep) and incubated overnight at 37˚ C, 5% CO$_2$. The following day, *T. cruzi* Tulahuen trypomastigotes (ATCC, Cat# 30266) were added to the flasks at a multiplicity of infection (MOI) of 5 and the flasks were returned to the incubator. After three days, the culture media was refreshed with fresh cMEM to provide optimal conditions for continued cell and parasite growth.

**Infection of mouse fibroblasts with *T. cruzi* trypomastigotes.** MC57G mouse fibroblasts (ATCC, Cat# CRL-2295) were cultured in cMEM media. To obtain enough peptides for LC-MS/MS analysis, a large quantity of infected cells was prepared to isolate the MHC-I complex from [13]. First, fibroblasts were seeded in five 500 cm$^2$ TripleFlask treated cell culture flasks (ThermoFisher Scientific, Cat# 132913) in 100 mL cMEM media and incubated at 37 ˚C, 5% CO$_2$. When the confluency reached >80%, fibroblasts were harvested from the flask using Accutase and counted using a Cellaca MX automated cell counter (Nexcelom). Then, 30 culture flasks of 175 cm$^2$ were prepared with 8 x 10$^6$ cells per flask in 30 mL cMEM media. After a 48-hour incubation at 37˚ C, fibroblasts in one flask were removed from the flask and counted. Based on the cell count, the number of parasites required to infect with a multiplicity of infection (MOI) of 7 was calculated.

To measure the MHC-I expression on *T. cruzi* infected MC57G fibroblasts, MC57G cells were seeded in 12-well plates in cMEM media and incubated overnight at 37 ˚C, 5% CO$_2$. The next day, T. cruzi trypomastigotes were added to the cells at an MOI of 7, and cells were further incubated for 48 hours. At the end of the incubation, extracellular parasites were washed off using 1x PBS, and cells were collected using Accutase. Detection of surface MHC-I was performed using an anti-mouse MHC-I Alexa Fluor 488 antibody (LS Bio, Cat# LS-C811400-50). Data was acquired using an Attune NxT flow cytometer (Thermofisher Scientific). Data was analyzed using FlowJo and median fluorescent intensity (MFI) was reported (S9 Fig).

**Preparation of lysate of *T. cruzi*-infected cells.** At the end of the 48-hour incubation of fibroblasts with *T. cruzi* trypomastigotes, cells were washed twice with PBS of 4 ˚C followed by detachment from the flask using a cell scraper. Scraped cells were collected in 1x PBS in 50-mL tubes and centrifuged for 10 min at 2000 x g at 4˚ C. Supernatant was discarded and pellets were either snap frozen in liquid nitrogen and stored at -80˚ C, or directly lysed.

Lysing the cells was performed by adding twice the volume of lysing buffer (0.5% Igepal CA-630 (Sigma-Aldrich, Cat# I8896), 150 mM NaCl, 50 mM Tris HCl pH 8, cOmplete mini protease inhibitor cocktail (Roche, Cat# 11836153001), and mass spectrometry-grade H$_2$O) to the cell pellet. After resuspending the pellet, the cells were incubated and rotated for 1 hour at 4˚ C. The lysed cells were then centrifuged for 10 min at 2000 x g to pellet their nuclei. The supernatant was transferred to a new tube and centrifuged for an additional 1 hour at 19,000 x g at 4˚ C. The supernatant was then filtered through a 0.45 µm and 0.2 µm filter to remove any possible particles that might clog the immunoaffinity purification column. Importantly, filters were prerinsed with in succession H$_2$O, methanol and again H$_2$O to remove plastic particles from the filter.

**Preparation of M1/42 monoclonal antibody.** M1/42 (M1/42.3.9.8) is a TIB-126 hybridoma-produced clone of an IgG2c rat monoclonal antibody that specifically binds to H-2 (MHC-I) haplotypes a, b, d, j, k, s, and u in mice [81]. TIB-126 hybridomas were purchased from ATCC and cultured in 30 mL complete IMDM media (Iscove's DMEM + 10% FBS (with IgG depleted (Thermofisher Scientific, Cat # 16250078) + 1x pen/strep) in 175 cm$^2$ non-

treated tissue culture flasks (Falcon, Cat# 355001) at 37 ˚C, 5% $CO_2$. Every 2–3 days, the viability of the cells was assessed and 3 mL of complete IMDM media was added. When the viability dropped below 50%, cell culture supernatant containing M1/42 was harvested by pelleting cells for 5 min at 400 x g. The supernatant was then filtered through a 0.45 μm filter and diluted with IgG binding buffer (Pierce, Cat# 21011) to a final pH of 5–5.5. The low pH is optimal for IgG binding to Protein G, and will improve the yield of the purification [82]. Using a peristaltic pump, the diluted supernatant was loaded onto a HiTrap protein G HP prepacked column (Cytiva, Cat# 45-000-053) that was pre-equilibrated with IgG binding buffer. Subsequently, the column was rinsed with IgG binding buffer followed by the elution of the bound M1/42 using pH 2.8 IgG elution buffer (Pierce, Cat# 21004). Tris-HCl pH 9 was added to the eluted M1/42 to establish a neutral pH, followed by dialysis using a 3,500 MWCO dialysis cassette (ThermoFisher Scientific, Cat# 66110) with 1x PBS pH 7.4 with multiple changes of dialysis buffer to remove traces of Tris. Finally, M1/42 monoclonal antibody (mAb) was analyzed by Coomassie SDS-PAGE to check for purity.

To test the functionality of the M1/42 antibodies, MC57G fibroblasts were first infected with *T. cruzi* Tulahuen trypomastigotes. After 48 hours, cells were rinsed with PBS and detached from flask using Accutase. Cells were incubated for 30 min at 4˚ C with M1/42 mAb purified from the TIB-126 hybridoma cell line, followed by ten washes with staining buffer (2% FBS in PBS) using the Laminar Wash HT2000 (Curiox Biosystems). Bound M1/42 to MC57G cells was then detected by a 30 min incubation with goat anti-mouse IgG PE (ThermoFisher Scientific, Cat# P-852) at 4 ˚C. After incubation, cells were washed 15 times, followed by analysis on a Guava Muse flow cytometer (Cytek).

**Immuno-affinity purification.** To prepare the immunoaffinity purification column, the previously purified M1/42 mAb was covalently linked to AminoLink Plus Coupling Resin (ThermoFisher Scientific, Cat#20501). Following the manufacturers recommendations, 5 mL of settled resin was incubated with 11.5 mg of M1/42 mAb in pH 7.2 1x PBS in a 15-mL tube. Cyanoborohydrate solution was added to a final concentration of 50 mM and resin was rotated overnight at 4˚ C. The next day resin was washed with coupling buffer followed by a wash with pH 7.5 quenching buffer (1M Tris HCl, pH 7.5). Next, resin was gently rocked for 30 min in 50 mM Cyanoborohydrate in quenching buffer. Then the resin was transferred to two 1.5 cm glass chromatography columns (Bio-Rad, Cat#7374150). After allowing it to settle, resin was washed with 1 M NaCl followed by a final wash with pH 7.2 coupling buffer with 0.05% sodium azide. Columns were then wrapped in aluminum foil to avoid light exposure and stored until use at 4˚ C.

Isolation and purification of MHC-I–peptide complexes was performed using an adapted method from Purcell and colleagues [83]. Mass spectrometry grade H2O (Optima, Cat# W64) was used to prepare all buffers which were freshly prepared. Briefly, the immunoaffinity purification column, consisting of M1/42 mAb covalently coupled to the AminoLink resin, was equilibrated with wash buffer 1 (0.005% Igepal CA-630, 150 mM NaCl, 50 mM Tris HCl pH 8, 5 mM EDTA, 100 μM PMSF, 1 μg/mL pepstatin A). Then cell lysate was passed over the column for four consecutive times, allowing all the MHC-I-peptide complexes to bind to the column. Next, a series of washes was performed; wash buffer 1 to remove unbound proteins, wash buffer 2 (150 mM NaCl, 50 mM Tris HCl pH 8) to remove detergent, wash buffer 3 (450 mM NaCl, 50 mM Tris HCl) to remove non-specific material, and last wash buffer 4 (50 mM Tris HCl) to remove salt. Finally, 10% acetic acid was used to elute the MHC-I-peptide complexes. The low pH of the elution buffer will unfold the MHC-I complex, resulting in elution of the peptides from the MHC-I binding cleft. Next, the peptides were separated from the unfolded MHC-I complex using a 10 kDa Amicon Ultra centrifugal filter (MilliPore Sigma, Cat# UFC801024) that was prerinsed with successively milliQ water, methanol, milliQ water,

and finally 10% acetic acid. The flow-through was concentrated to 100–250 μL using a speed vac and then stored at -80˚ C for LC-MS/MS analysis.

**Mass spectrometry.**   Peptides were cleaned up using a HyperSep C18 cartridge (Thermo 60108–376). Cartridges were rinsed with 500 μL of 80% Acetonitrile 0.1% formic acid, then with 500 μL 50% acetonitrile 0.1% formic acid, and equilibrated with 1 mL of LCMS grade water in 0.1% formic acid. Immunopeptides were loaded onto the cartridge and the cartridge washed with 500 μL of the 0.1% formic acid solution. Peptides were eluted with 300 μl of 50% acetonitrile, 0.1% formic acid and then dried in a speed vac and resuspended in 30 μL of 2% acetonitrile, 0.1% formic acid, 97.9% water and placed in an autosampler vial.

Samples were analyzed by nanoLC-MS/MS (nanoRSLC, ThermoFisher) using an Aurora series (Ion Opticks) reversed phase HPLC column (25 cm length x 75 μm inner diameter) directly injected into a ThermoScientific Orbitrap Eclipse using a 160 min method (mobile phase A = 0.1% formic acid (Thermo Fisher), mobile phase B = 99.9% acetonitrile with 0.1% formic acid (Thermo Fisher); hold 3% B for 15 min, 3–22% B in 95 min, 22–38% for 24 min, 38%-90% for 2min, hold at 90% for 2min, 90–5% in 2 min, followed by a second quick gradient and equilibration) at a flow rate of 300 μL/min. Eluted peptide ions were analyzed using a data-dependent acquisition (DDA) method with resolution settings of 120,000 and 7,500 (at m/z 200) for MS1 and MS2 scans, respectively. DDA-selected peptides were fragmented using high energy collisional dissociation (30%). Raw data files are made publicly available on the Dryad repository [84].

PEAKS version X software was used to process all data dependent acquisition mass-spectral data [85]. Proteins identifications were obtained by searching a database of *Mus Musculus* C57BL/6J obtained from Uniprot (6th October 2017) and a database of *Trypanosoma cruzi* CL Brener obtained from Uniprot (23rd October 2021, UP000002296). The following data analysis parameters were used: enzyme set to none, digest mode set to unspecific, up to three variable modifications including oxidation of methionine as well as deamidation of asparagine and glutamine, parent mass error tolerance of 15 ppm, fragment mass error tolerance of 0.02 Da, charge states between 1 and 7 were accepted, the peptide -10LgP score was left at the default 15 and the protein -10LgP score was left at the default 20 corresponding to protein false discovery rates of 1.6% and 2.5%. Peptide sequence lengths between 8 and 15 amino acids were selected for further analysis.

To verify the accuracy of the mass spectrometry spectral sequence annotation of the Tcj2 derived peptides, the peptides LNIKHLDDRDVS, VKETKFYDSLG, AFYTGKTIKLA, LEAFYTGKTIKLA, SNEISDLR, and GEGDQIPGVR were synthesized and analyzed using identical settings by nanoLC-MS/MS (nanoRSLC, ThermoFisher). The MS spectra from these synthetic peptides were compared to the Tcj2-identified peptides in the T. cruzi–infected MC57G fibroblasts to confirm their identity.

## Protein sequence alignments

To compare the Tcj2 protein sequence between different strains of *T. cruzi*, amino acid sequences of ortholog syntenic genes of Tcj2 of the following strains were downloaded from TriTrypDB.org (accessed on May 23rd, 2023): Dm28c (C4B63_175g9), Sylvio X10/1 (TCSYLVIO_003497), G (TcG_07973), Brazil A4 (TcBrA4_0100200), Berenice (ECC02_009221), Y C6 (TcYC6_0075780), CL Brener Esmeraldo-like (TcCLB.511627.110), TCC (C3747_88g53), and CL (TcCL_ESM07472). Multiple sequence alignments were done using Clustal Omega, and the results were ordered by The Distinct Types Units (DTUs) genetic classification each strain belongs to [86].

Additionally, homologous DnaJ protein sequences of *Trypanosoma rangeli* (TRSC58_00469), *Trypanosoma brucei brucei* (Tb927.2.5160), *Trypanosoma congolense* (TcIL3000_2_1270) and *Trypanosoma vivax* (TvY486_0007950) were also compared to *T. cruzi* Tcj2 by multiple sequence alignment using Clustal Omega.

Comparison of DnaJ-homolog protein sequences between species *Trypanosoma cruzi*, *Mus musculus* and *Homo sapiens* were also performed. *T. cruzi* Tcj2 (Uniprot: Q4D832) was ran through protein BLAST (https://blast.ncbi.nlm.nih.gov) and results were filtered for *M. musculus* or *H. sapiens*. From the significant alignment results list, the sequence with the best score (lowest E value) was selected, resulting in DnaJ homolog subfamily A member 4 isoform 2 (NP_067397.1) for *M. musculus* and DnaJ homolog subfamily A member 4 isoform 2 (NP_001123654.1) for *H. sapiens*. Using Clustal Omega the multiple sequence alignment was performed.

## mRNA vaccine construct encoding *T. cruzi* Tcj2

A mRNA vaccine construct encoding for *T. cruzi* Tcj2 (Uniprot: Q4D832) was ordered from the RNAcore of Houston Methodist Research Institute. At the C-terminus of the Tcj2 protein, the amino acid sequence SIINFEKL was added, followed by the sequence for a FLAG-tag. The complete sequence was optimized for improved RNA translation. Uridine was replaced by N1-Methylpseudouridine during the *in vitro* transcription of the mRNA to achieve improved translation and vaccine efficacy [87,88]. mRNA was capped using CleanCap (Trilink Technology). An automated electrophoresis gel (Tapestation, Agilent) was ran by the RNAcore, to verify the study integrity of the Tcj2 mRNA (S10 Fig).

## *In vitro* testing of Tcj2 mRNA construct

**MHC-I presentation.**   The DC2.4 murine dendritic cell line (Millipore Sigma, Cat# SCC142) was cultured in DC2.4 media (RPMI 1640 + L-glutamine, 10% FBS, 1x pen/strep, 1x non-essential amino acids, 10 mM HEPES, 55 µM β-mercaptoethanol) and used for *in vitro* transfections. DC2.4 cells were seeded at 240,000 per well in 12-well culture plates and incubated overnight at 37° C. The following day, cells were transfected with 1 µg Tcj2 mRNA in 2 µL Lipofectamine MessengerMAX (Invitrogen, cat# LMRNA001) and Opti-MEM (Gibco, Cat# 31985062) per well according to manufacturer's instructions. As controls, cells were transfected with 1 µg of ovalbumin expressing mRNA (5MoU modified, TriLink, Cat# L-7210) in 2 µL Lipofectamine MessengerMAX or with transfection agent only. After 24 hours incubation at 37 °C, transfected cells were harvested by a 15-minute incubation with Accutase (Sigma-Aldrich, Cat# A6964) followed by the use of a cell scraper. Cells were washed using DC2.4 media and centrifugation, and cell viability and concentration was measured using ViaStain AO/PI viability dye (Nexcelom, Cat# CS2-0106) and the Cellaca MX automated cell counter (Nexcelom, PerkinElmer).

**Intracellular detection of FLAG-tag.**   The expression of Tcj2 protein after mRNA transfections was assessed by flow cytometry using intracellular staining of the FLAG-tag incorporated in the Tcj2 mRNA construct. For all the washing steps in this protocol the Laminar Wash HT2000 (Curiox Biosystems) was used for improved cell viability and recovery. Briefly, 100,000 transfected cells were added per well in a laminar wash 96-well plate (Curiox, Cat# 96-DC-CL-05) and the plate was incubated for 20 min at 4° C. After the cells were settled, ten washes with staining buffer (2% FBS in PBS) were performed and cells were resuspended in Cytofix/Cytoperm (BD, Cat# 51-2090KZ). After a 20 min incubation at 4 °C, cells were washed 10x in 1x Perm/Wash buffer (BD, Cat# 51-2091KZ). Then, cells were resuspended in 25 µL 1x Perm/Wash buffer and 1 µL of mouse Fc Block (BD, Cat# 553142) was added and cells were

incubated for 5 min at 4˚ C. Subsequently, anti-FLAG M2-Cy3 antibody (Sigma-Aldrich, Cat# A9594) was diluted 1:100 in 1x Fix/Perm buffer and 45 μL was added to the cells. Following a 30 min incubation at 4˚ C, cells were washed 15x in staining buffer. Cells were then transferred from the laminar wash plate to 1.5 mL Eppendorf tubes and analyzed for intracellular FLAG-staining using a Guava Muse flow cytometer (Cytek).

**Presentation of SIINFEKL peptide on MHC-I.** To assess MHC-I presentation of peptides derived from the translated Tcj2 mRNA, DC2.4 cells were stained with fluorophore conjugated antibody that recognizes SIINFEKL peptide bound to H2-K$^b$. To each well of a laminar wash 96-well plate (Curiox, Cat# 96-DC-CL-05) 100,000 transfected cells were added and the plate was incubated for 20 min at 4 ˚C. After the cells were settled, ten washes with staining buffer (2% FBS in PBS) were performed using the Laminar Wash HT2000. Cells were then resuspended and 1 μL of mouse Fc Block was added and cells were incubated for 5 min at 4 ˚C, followed by the addition of 0.625 μL PE anti-mouse H-2K$^b$ bound to SIINFEKL Antibody (Biolegend, Cat# 141604). After a 30 min incubation at 4 ˚C, cells were washed 15x in staining buffer. Cells were then transferred from laminar wash plate to 1.7 mL Eppendorf tubes and analyzed for intracellular FLAG using a Guava Muse flow cytometer (Cytek).

**Activation of SIINFEKL-specific T cells.** *In vitro* co-culture experiments were conducted to examine the activation of naïve CD8+ T cells by mRNA transfected cells by measuring the cytokine secretions in the supernatant 24 hours after the start of the co-culture. DC2.4 cells were cultured and transfected with Tcj2 mRNA, Ovalbumin mRNA (positive control), or only the transfection agent Lipofectamine MessengerMAX (negative control) as earlier described. The following day cells were harvested using Accutase and counted using the Cellaca MX automated cell counter. Transfected DC2.4 cells were then seeded in 96-well culture plates at 75,000 cells per well in DC2.4 media and incubated at 37 ˚C, 5% CO$_2$. In the meantime, splenocytes from naïve C57BL/6J mice or naïve C57BL/6J OT-1 mice were thawed from -150 ˚C in DC2.4 media and counted. Five hours after the transfected DC2.4 cells were seeded, 750,000 C57BL/6J or C57BL/6J OT-1 splenocytes were added to designated wells and incubation was continued overnight. After 24 hours, the 96-well culture plate was centrifuged for 5 min at 300 x g to pellet cells, and supernatant was harvested and frozen at -80 ˚C. To analyze the cytokines that were secreted during the co-culture, a Luminex cytokine assay was performed on the culture supernatants. A MILLIPLEX MAP Mouse CD8+ T Cell Magnetic Bead Panel kit (Millipore Sigma, Cat# MCD8MAG-48K) containing the analytes Granzyme B, IFN-γ, TNF-α, IL-2, IL-6 and IL-4 was used in combination DropArray technology according to a previous published method [89].

## mRNA vaccine formulation in Lipid Nanoparticles (LNPs)

Tcj2 mRNA LPNs were prepared using the Genvoy ILM kit (Precision Nanosystems, Cat# NWW0042) according to the manufacturer's recommendation. The kit contains PNI ionizable lipid, DSPC, Cholesterol and PNI stabilizer at a mol% of 50, 10, 37.5 and 2.5, respectively. The Tcj2 mRNA LNPs were prepared at N:P ratio (nitrogen to phosphate) of 4:1 and formulated using a NanoAssmblr Ignite (Precision Nanosystems) instrument. LNPs without mRNA were prepared as controls (hence called empty LNPs). After formulation, LNPs were concentrated using 10 kDa spin filter columns and 0.2 μm sterile filtered. Using a RiboGreen RNA Assay Kit (Invitrogen, Cat# R11490) and 1x TE buffer with and without TritonX100 detergent, the RNA concentration in the LNPs was calculated. The LNPs were then diluted in 1x PBS with a final concentration of 8% sucrose to increase their stability during freezing [79]. These vaccine formulations were stored at -80 ˚C until use.

To characterize the LNPs, average LNP size and polydispersity index was determined by dynamic light scattering (DLS) using a DynaPro Plate Reader II instrument (Wyatt). Samples were diluted in PBS prior to testing.

To measure the pKa (charge) of the surface of the LNPs, a 2-(p-toluidino)-6-naphthalene sulfonic acid (TNS) fluorescent assay was conducted according to methods published by Patel *et al* [49]. Data was analyzed using Prism 9 and a four-parameter dose-response curve was fitted to the data points to obtain the pKa ($IC_{50}$ of the data).

## Production of recombinant Tcj2 protein

To produce recombinant Tcj2 protein, a pET41 a(+) expression vector was designed containing the *T. cruzi* Tcj2 sequence (Uniprot: Q4D832) followed by a His6-tag at the C-terminus. The sequence was codon optimized for protein expression in *E. coli* (Genscript). For protein expression, *E. coli* BL21 cells containing Tcj2-pET41a vector were cultured in LB media at 37˚ C in the presence of kanamycin to an O.D. of 0.6. Subsequently, protein expression was induced with 0.5 mM IPTG at 22˚ C for 4 hours. Next, *E. coli* cells were harvested by centrifugation followed by cell lysis of the pellet using Bugbuster protein extraction reagent (Millipore, Cat# 70584). The recombinant Tcj2 protein was initially purified by Immobilized Metal Affinity Chromatography (IMAC) using the HisTrap FF column (Cytiva, Cat# 17525501). The sample was applied to this column after first adding 2M urea. The column was washed with IMAC buffer (30 mM Tris-HCl pH 7.5, 500 mM NaCl) with 2 M urea and bound protein was refolded using a linear gradient of 2–0 M urea in IMAC buffer. After further washing the column with IMAC buffer with 20 mM imidazole, bound protein was eluted using a linear gradient of 0–500 mM imidazole in IMAC buffer. Then, the protein was further purified using a Butyl Sepharose column (Cytiva, Cat# 28411001). Therefore, ammonium sulfate salt was added to a final 1 M concentration to the IMAC purified protein. The column was washed with HIC buffer (20 mM Tris-HCl pH 8) with 1 M ammonium sulfate and bound protein was eluted in a linear gradient of 1–0 M ammonium sulfate in HIC buffer. Finally, removal of endotoxin was done by incubating the purified rTcj2 protein with Triton X-114, followed by Triton elimination using SM2 beads as earlier described [90]. After buffer exchange with 1x PBS by dialysis, final rTcj2 protein was stored at -80˚ C, and aliquots of rTcj2 were ran by SDS-PAGE followed by Coomassie staining or western blotting detecting HIS-tag using AP-conjugated anti-His tag antibody (ThermoFisher Scientific, Cat# R932-25).

## *In vivo* mouse immunogenicity study

For the first *in vivo* immunogenicity study, ten female C57BL/6J mice (The Jackson Laboratory) were used. To show reproducibility, a second set of female C57BL/6J mice was used at a later timepoint. Both *in vivo* studies were executed identically. At 6–8 weeks of age, five mice were immunized with 10 μg Tcj2 mRNA formulated in LNPs. Five other mice received empty LNPs as a negative control. Immunizations were administered intramuscular in the hind muscle with a volume of 50 μL. Twenty-one days after the first immunization, mice received booster immunization in an identical way as described for the primary immunization and were euthanized at day 40. Throughout the study all animals appeared clinically normal.

**Processing sera and spleens.** At day 40, mice were anesthetized by an intraperitoneal injection with ketamine/xylazine. Blood was collected through a cardiac puncture, followed by the harvest of the spleen. Sera was prepared by allowing the blood to clot in Z-Gel sera collection tubes (Sarstedt, Cat# 101093–958), followed by centrifugation for 5 min at 10,000 x g. The sera were transferred to new tubes and stored at—80˚ C until further use. For processing the spleens to single cell suspensions, all steps were performed at 4 ˚C or on ice. First, spleens were

rinsed in 1x PBS, transferred to gentleMACS C tubes (Miltenyi Biotec, Cat# 130-093-237) and then dissociated to a single cell suspension using the gentleMACS Dissociator (Miltenyi Biotec). After pelleting the splenocytes by centrifugation for 5 min at 300 x g, 1 mL ACK lysing buffer was added to lyse the red blood cells. After 1 min incubation on ice, 20 mL 1x PBS was added to stop the lysing reaction and splenocytes were centrifuged another time. Subsequently, supernatant was discarded and splenocytes were resuspended in complete RPMI (cRPMI) consisting of RPMI 1640 with L-Glutamine, 10% heat-inactivated fetal Bovine serum (FBS) and 1x pen/strep. Splenocytes were passed through a 40 μm strainer, and viability and concentration were assessed using a Cellaca MX automated cell counter (Nexcelom, PerkinElmer) and ViaStain AO/PI viability dye (Nexcelom, Cat# CS2-0106). Splenocyte suspensions were stored at 4°C until further use.

**SIINFEKL-specific CD8+ T cell analysis.** Splenocytes were washed once in 1x PBS and transferred to a laminar wash plate. After a 30 min incubation at 4°C to settle the cells, the laminar wash plate was washed 10x using the Laminar Wash HT2000 (Curiox Biosystems). Splenocytes were resuspended in 1:1000 diluted LIVE/DEAD Fixable Near-IR Dead Cell Stain (Invitrogen, Cat# L34975) and incubated at 4°C. After 30 min, the laminar wash plate was washed for 10x with staining buffer (2% FBS in 1x PBS) followed by the addition of mouse Fc block (BD, Cat# 553141). After a 5 min incubation, splenocytes were stained with Pacific Blue anti-mouse CD3ε (Biolegend, Cat# 100334), PerCP-Cy5.5 Anti-Mouse CD8a (BD, Cat# 551162) and PE labeled SIINFEKL-loaded mouse H-2K$^b$ tetramer (MHC Tetramer Production Facility, Baylor College of Medicine). After a 30 min incubation at 4°C, cells were washed 10x with staining buffer followed by addition of BD Cytofix buffer (BD, Cat# 554655) to fix the cells. After 30 min at 4°C, cells were washed 15x with staining buffer. Cells were analyzed using an Attune NxT flow cytometer (ThermoFisher Scientific) and data was analyzed using VenturiOne software V6 (Applied Cytometry). Gating strategy can be observed in S11 Fig.

**Enzyme-Linked Immunosorbent Assay (ELISA) detecting anti-Tcj2 antibodies.** Indirect ELISAs were performed to assess Tcj2-specific antibody titers of total IgG, IgG1 and IgG2c. The 96-well ELISA plates were coated overnight at 4°C with 0.25 μg/mL rTcj2 diluted in KPL coating solution (SeraCare, Cat# 5150–0014). The following day the coating solution was discarded, and wells were blocked for two hours at room temperature with dilution buffer (0.1% Bovine Serum Albumin (BSA) in 1x PBS + 0.05% Tween-20 (PBST)). Mouse serum was serially diluted two-fold in dilution buffer, starting at 1:200. As a negative control, a pool of naïve sera was diluted at 1:200. After blocking, dilution buffer in wells was discarded, wells were washed once using a BioTek 405TS plate washer and PBST, and 100 μL of diluted sera or negative control pooled sera was added to designated wells in duplicate. After a two-hour incubation at room temperature, the plate was washed four times, followed by the addition of 100 μL/well goat anti-mouse IgG HRP (Lifespan Bioscience), goat anti-mouse IgG1 HRP (Lifespan Bioscience), or goat anti-mouse IgG2c HRP (Lifespan Bioscience). After one hour of incubation at room temperature, the plates were washed five times with PBST followed by a 15 min incubation with 100 μL/well TMB substrate (KPL). The color reaction was stopped by the addition of 100 μL/well 1M HCl, and the absorbance at 450 nm was measured using a spectrophotometer (Epoch 2, BioTek). To analyze the results, the $OD_{450}$ values from duplicate wells were first averaged. Then the antibody titer cutoff value was calculated using the formula: average of naïve sera control + 3 x standard deviation of the naïve sera control. Represented are end-point antibody titers, defined as highest serum dilution that still resulted in an $OD_{450}$ value above the cutoff value. When a sample did not show any signal at all and the antibody titer could not be calculated, an arbitrary baseline antibody titer value of 67 was assigned.

**Native Tcj2 western blots.** Western blots were performed to analyze Tcj2 in *T. cruzi* lysate, as well as testing for cross-reactivity of homologous-DnaJ protein in lysates from

MC57G mouse fibroblasts and HEK293T human kidney cells. *T. cruzi* Tulahuen lysate was prepared according to previous published methods [91]. For lysates from MC57G and HEK293T cells, cells cultured in flasks were washed three times with 1x PBS to remove culture media and cells were detached from the flask using a cell scraper. Cells were collected by centrifugation and resuspended in 500 μL RIPA buffer (ThermoFisher Scientific, Cat# 89900) and incubated on ice for 15 min. Then, lysed cells were centrifuged for 15 min at 12,000 x g and the supernatant was collected and filtered through a 0.2 μm filter. Protein concentration was quantified using a BCA protein quantification kit (ThermoFisher Scientific, Cat# 23225) according to manufacturer's recommendations.

Cell lysate were loaded on a 4–12% Bis-Tris SDS-PAGE gel, using rTcj2 protein as a positive control and Bovine Serum Albumin (BSA) as a negative control. The SDS-PAGE gel was run for 75 min at 140 V, followed by blotting of the proteins to nitrocellulose. The blot was incubated overnight at 4° C with pooled Tcj2 antisera diluted 1:5000 in 1% non-fat dry milk in PBST. Goat anti-mouse IgG alkaline phosphatase (KPL, Cat# 5220–0357) diluted 1:5000 in PBST was used as a secondary detection antibody. For confirmation of human DNAJA4 in HEK293T cell lysate, an anti-human DNAJA4 Polyclonal Antibody (ThermoFisher Scientific, Cat# PA5-65311) was used at a 1:500 dilution and in combination with ECL substrate (Cytiva, Cat# RPN2236).

**Analysis of antigen-specific T cells by flow cytometry.** To assess the induction of Tcj2 specific T cells by the mRNA vaccine, $1 \times 10^6$ live splenocytes were seeded per well in a 96-well culture plate and restimulated with 10 μg/mL rTcj2 protein in cRPMI media *in vitro*. Unstimulated (negative) and PMA/I (positive) stimulated controls were included for each splenocyte sample. Splenocytes were incubated for 48 hours at 37° C, 5% $CO_2$, with the last 5 hours in presence of Brefeldin A (BD Biosciences, Cat# 555029) to retain cytokines intracellularly. After this incubation, cells were transferred to a laminar wash 96-well plate (Curiox, Cat# 96-DC-CL-05) and the plate was incubated for 20 min at 4° C. For all the washing steps in this protocol the Laminar Wash HT2000 (Curiox Biosystems) was used for improved cell viability and recovery. After the cells were settled, ten washes with 1x PBS were performed and cells were resuspended in viability dye (Fig 9). After a 30 min incubation at 4° C, ten washes with staining buffer (2% FBS in PBS) were performed. Then the Fc receptors CD16/CD32 on the cell surface were blocked with 2 μL mouse Fc Block (BD, Cat# 553142) per well for 5 min, followed by the addition of the surface marker antibody cocktail (containing CD3, CD4, CD8, CD19, CD25, CD44, CD62L, CD127 and TCRγδ, Fig 9). After a 30 min incubation at 4° C, ten washes with staining buffer (2% FBS in PBS) were performed and cells were resuspended in Cytofix/Cytoperm (BD, Cat# 51-2090KZ). After a 20 min incubation at 4° C, cells were washed 10x in 1x Perm/Wash buffer (BD, Cat# 51-2091KZ), followed by the addition of the intracellular marker antibody cocktail (containing IL-17A, Granzyme B, IL-10, IFN-γ, TNF-α and perforin, Fig 9). Following a 30 min incubation at 4° C, cells were washed 15x in staining buffer. Cells were then transferred from laminar wash plate to 96-well culture plates and analyzed using an Aurora spectral flow cytometer (Cytek). To unmix the raw data, single stained cell and bead controls were used. Further analysis of flow cytometry data was done using FlowJo software. Gating of cell populations was done using Fluorescence minus one (FMO) samples and untreated controls (S12 Fig). The %-values of populations from unstimulated cells were subtracted from %-values from rTcj2-stimulated cells to obtain the antigen-specific results. A Boolean gating strategy was applied to analyze the polyfunctional CD8+ T cells producing two or more cytokines. Boolean combination gates were drawn in the FlowJo software.

**In vitro *T. cruzi*—Infected fibroblast killing assay.** To measure the cytotoxic effector functions of CD8+ T cells from immunized mice, a co-culture between *T. cruzi* infected MC57G fibroblasts and splenocytes was conducted. First, 15,000 MC57G murine fibroblasts

| Target | Fluorochrome | Clone mAb | Manufacturer | Catalogue # |
| --- | --- | --- | --- | --- |
| CD3 | APC/Fire 810 | 17A2 | Biolegend | 100268 |
| CD4 | BV605 | RM4-5 | BD Biosciences | 563151 |
| CD8a | BUV615 | 53-6.7 | BD Biosciences | 613004 |
| CD19 | BV480 | 1D3 | BD Biosciences | 566167 |
| CD25 | BB790 | PC61 | BD Biosciences | 624296 |
| CD44 | BV570 | IM7 | BD Biosciences | 624298 |
| CD62L | APC-Cy7 | MEL-14 | BD Biosciences | 560514 |
| TCRγδ | PE-Cy5 | eBioGL3 | Life technologies | 15-5711-82 |
| IL-17A | BV786 | TC11-18H10 | BD Biosciences | 564171 |
| Granzyme B | BV421 | GB11 | BD Biosciences | 563389 |
| IFN-γ | PE | XMG1.2 | Biolegend | 505808 |
| TNF-α | BV510 | MP6-XT22 | Biolegend | 506339 |
| Perforin | FITC | S16009A | Biolegend | 154310 |
| Viability | ViaDye™ Red Fixable Viability Dye | N/A | Cytek | R7-60008 |

**Fig 9. Flow cytometry fluorophore-conjugated antibodies and viability dye used for the experiment.**

were seeded in 96-well flat-bottom culture plates and incubated overnight at 37 ˚C, 5% $CO_2$. The next day, 150,000 β-galactosidase-expressing *T. cruzi* trypomastigotes (Tulahuen, *LacZ*) (BEI Resources, Cat# NR-18959) were added to the fibroblasts and culture plates were further incubated at 37 ˚C, 5% $CO_2$. After 24 hours, media in wells was removed with a multichannel pipet and wells were washed twice with 1x PBS to remove extracellular *T. cruzi* trypomastigotes. Subsequently, 1 x $10^6$ live splenocytes were added to designated wells in complete co-culture media (RPMI 1640 + L-glutamine, 10% FBS, 1x pen/strep, 1x non-essential amino acids, 10 mM HEPES, 55 μM β-mercaptoethanol). In addition, naïve C57BL/6J were added to designated wells as negative controls, as well as 100 μM benznidazole (Sigma, Cat# 419656-1G) as positive control for *T. cruzi* killing. In addition to *T. cruzi*–infected fibroblasts, a similar co-culture was prepared with non-infected fibroblasts. Finally, the co-culture was then incubated for 72 hours. Afterwards, supernatant was harvested for cytokine analysis, and cells were incubated with 1% Igepal-630 in PBS supplemented with 100 μM Chlorophenol Red-β-D-galactopyranoside. After a 4-hour incubation at 37 ˚C to lyse the cells and convert the substrate, the substrate conversion by β-galactosidase was analyzed by measuring the absorbance at 570 nm using a spectrophotometer (Epoch 2, BioTek). For analysis of the results, $OD_{570}$ values of co-cultures without *T. cruzi*–infected fibroblasts were first subtracted from $OD_{570}$ values from co-cultures with *T. cruzi*–infected fibroblasts to remove background. Then, the $OD_{570}$ values were normalized between maximum killing of intracellular *T. cruzi* parasites (co-culture with 100 μM benznidazole) and no killing of intracellular *T. cruzi* parasites (co-culture with naïve C57BL/6J splenocytes).

**Data analysis.** All data was plotted with Prism 9 (GraphPad) and analyzed for statistical significance using a non-parametric Mann-Whitney U test. Stars representing statistical significance indicate the following: *: $p < 0.05$, **: $p < 0.01$, ***: $p < 0.001$, ****: $p < 0.0001$.

## Supporting information

**S1 Table. List of all the T. cruzi peptides presented on MHC-I.** Properties of each peptide calculated by mass spectrometry are listed. Affinity (in nM) to H2-Kb and H2-Db was calculated using NetMHCpan 4.1 prediction tool.
(TIF)

**S1 Fig. Amino acid sequence of Tcj2 protein is very conserved between different Discrete Typing Units (DTUs) of *T. cruzi*.** Multiple sequence alignment of DnaJ (Tcj2) protein sequence of different *T. cruzi* strains, obtained from TriTripDB.org (accessed on May 23[rd], 2023). The two mutations in amino acid residues are highlighted in red.
(TIF)

**S2 Fig. Multiple Sequence Alignment (Clustal Omega) between Tcj2 from *Trypanosoma cruzi* and DnaJ homolog from *Mus musculus* and *Homo sapiens*.** The colored sequences represent the six peptides that were found by immunopeptidomics. Sequence identity is 42% between *T. cruzi* and *M. musculus* (dnaJ homolog subfamily A member 4 isoform 2). Sequence identity is 43% between *T. cruzi* and *H. sapiens* (dnaJ homolog subfamily A member 4 isoform 2).
(TIF)

**S3 Fig. Spectral validation of Tcj2 peptides from immunopeptidomics experiment.** Spectra from Tcj2 reference synthetic peptides were compared to spectra from experimental peptides, and similarities confirmed the identity of these peptides. Corresponding -10LogP values for the experimental peptides were deemed to be significant as determined by the PEAKS algorithm to be 21.4, 30.1, 28.3, 20.7, 17.3, 30.2 from top to bottom. The measured mass error was less than 4 ppm for all peptides.
(TIF)

**S4 Fig. Multiple Sequence Alignment (Clustal Omega) between Tcj2 from *Trypanosoma cruzi* and Tbj2 from *Trypanosoma brucei brucei*.** Protein sequence obtained from TriTripDB.org (accessed on July 3rd, 2023).
(TIF)

**S5 Fig. Recombinant *T. cruzi* Tcj2 was expressed in *E. coli* and purified.** Coomassie stained SDS-PAGE gel and western blot detecting HIS-tag showed a main band at the expected size of 45 kDa. Three bands smaller than the main band were observed, of which one band contained the HIS-tag. Densitometry analysis estimated the purity of the main band to be 76%.
(TIF)

**S6 Fig. Tcj2 antisera recognizes native Tcj2 in *T. cruzi* lysate, but not DnaJ-homologs in lysate from *H. sapiens*, *M. musculus* and *T. brucei*.** 20 ng rTcj2 protein, as well as 3 μg of lysates from *T. cruzi* Tulahuen, *H. sapiens* (HEK293T), *M. musculus* (MC57G) and *T. brucei brucei*, were ran on reduced SDS-PAGE gels and either stained with Coomassie Blue or subjected to western blotting followed by incubation with Tcj2 antisera from Tcj2 LNP-vaccinated mice. 20 ng Bovine Serum Albumin (BSA) protein was added as negative control.
(TIF)

**S7 Fig. Western blot confirming the presence of human DNAJA4 in HEK292T cell lysate.**
100 ng rTcj2 protein, as well as 3 μg of lysates from *T. cruzi* Tulahuen, *H. sapiens* (HEK293T) and *M. musculus* (MC57G) were ran on reduced SDS-PAGE gels and either stained with Coomassie Blue or subjected to western blotting followed by incubation with anti-human DNAJA4 polyclonal antibody.
(TIF)

**S8 Fig. Cytokines produced by CD4+ T cells after *in vitro* restimulation with rTcj2 protein.**
No significant changes in cytokine production were observed by CD4+ T cells after restimulation, but an observable trend in increase in IFN-γ was observed. Data values from non-stimulated cells were subtracted from rTcj2 protein stimulated cells to obtain antigen-specific cytokine production. Mean and standard deviation are shown. Filled symbol shapes represent immunogenicity study #1, while open symbol shapes represent repeat immunogenicity study #2.
(TIF)

**S9 Fig. Histogram plot showing the expression of MHC-I by *T. cruzi* infected and noninfected MC57G mouse fibroblasts.** Cells were infected with *T. cruzi* for 48 hours followed by flow cytometric staining. As a positive control for upregulated MHC-I expression, cells were incubated for 24 hours with recombinant mouse IFN-γ. VERO cells that do not have mouse MHC-I were used as a negative control.
(TIF)

**S10 Fig. Electrophoresis gel showing the robust integrity of Tcj2 mRNA.** Tapestation automated electrophoresis was performed to analyze the size and integrity of the Tcj2 mRNA construct. A strong single band was observed, calculated to be 1404 nucleotides, which was the expected size. The second band at 25 nucleotides represents a positive control that is run within the sample.
(TIF)

**S11 Fig. Flow cytometry gating strategy to analyze SIINFEKL-specific CD8+ T cells.** After lymphocytes and singlets were selected, live CD3+ T cells were gated, followed by CD8+ T cells, followed by gating on SIINFEKL tetramer PE positive cells.
(TIF)

**S12 Fig. Flow cytometry gating strategy to analyze *in vitro* rTcj2-restimulated splenocytes.**
First live T cells were gated, followed by gating on CD4+, CD8+ or γδ T cells. CD8+ T cells were further separated by central memory or effector memory CD8+ T cells. For all cell populations intracellular cytokine production was measured. Additionally, CD8+ T cells were also analyzed for polyfunctionality, meaning the intracellular production of two or more cytokines.
(TIF)

## Acknowledgments

The authors would like to extend their sincere appreciation to Dr. Ziyin Li from the Department of Microbiology and Molecular Genetics at UT Health for generously providing the *T. brucei brucei* lysate. Additionally, we would like to express our heartfelt gratitude to all the scientists from the Houston Methodist RNA Core for their invaluable contributions and support in the development and generation of our mRNA. This project was also supported by the Cytometry and Cell Sorting Core at Baylor College of Medicine with funding from the NIH (NIAID P30AI036211, NCI P30CA125123, and NCRR S10RR024574) and the assistance of

Joel M. Sederstrom. The LC-MS/MS analysis was conducted at The University of Texas Medical Branch's Mass Spectrometry Facility and supported by the Cancer Prevention Research Institute of Texas (CPRIT) grant no. RP190682.

## Author Contributions

**Conceptualization:** Leroy Versteeg, Edwin Tijhaar, Jeroen Pollet.

**Data curation:** Leroy Versteeg, William K. Russell.

**Formal analysis:** Leroy Versteeg, Edwin Tijhaar, Jeroen Pollet.

**Funding acquisition:** Kathryn Jones, Maria Elena Bottazzi, Peter Hotez, Jeroen Pollet.

**Investigation:** Leroy Versteeg, Rakesh Adhikari, Gonteria Robinson, Jungsoon Lee, Junfei Wei, Nelufa Islam, Brian Keegan, William K. Russell, Cristina Poveda, Maria Jose Villar, Jeroen Pollet.

**Methodology:** Leroy Versteeg, Rakesh Adhikari, Edwin Tijhaar, Jeroen Pollet.

**Project administration:** Leroy Versteeg.

**Resources:** Kathryn Jones, Jeroen Pollet.

**Supervision:** Leroy Versteeg, Kathryn Jones, Maria Elena Bottazzi, Peter Hotez, Edwin Tijhaar, Jeroen Pollet.

**Validation:** Leroy Versteeg, Edwin Tijhaar, Jeroen Pollet.

**Visualization:** Leroy Versteeg, Edwin Tijhaar.

**Writing – original draft:** Leroy Versteeg, William K. Russell, Edwin Tijhaar, Jeroen Pollet.

**Writing – review & editing:** Leroy Versteeg, Rakesh Adhikari, Jungsoon Lee, Nelufa Islam, Brian Keegan, William K. Russell, Cristina Poveda, Maria Jose Villar, Kathryn Jones, Maria Elena Bottazzi, Peter Hotez, Edwin Tijhaar, Jeroen Pollet.

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
