## [Decision Letter · Decision Letter 0]

18 Feb 2024

Dear MR Versteeg,

Thank you very much for submitting your manuscript "Immunopeptidomic MHC-I Profiling and Immunogenicity Testing Identifies Tcj2 as a New Chagas Disease mRNA Vaccine Candidate." for consideration at PLOS Pathogens. As with all papers reviewed by the journal, your manuscript was reviewed by members of the editorial board and by several independent reviewers. In light of the reviews (below this email), we would like to invite the resubmission of a significantly-revised version that takes into account the reviewers' comments.

Reviewers generally agree that the study could be highly impactful. And this is why it is important that it is well controlled. Aside from the "smaller scale" comments provided by reviewers 2+3, the issues raised by reviewer 1 (who is an immunopeptidomics expert) are important and should be fully addressed.

We cannot make any decision about publication until we have seen the revised manuscript and your response to the reviewers' comments. Your revised manuscript is also likely to be sent to reviewers for further evaluation.

Sincerely,

F. Nina Papavasiliou

Academic Editor

PLOS Pathogens

Jeffrey Dvorin

Section Editor

PLOS Pathogens

Michael Malim

Editor-in-Chief

PLOS Pathogens

orcid.org/0000-0002-7699-2064

Reviewers generally agree that the study could be highly impactful. And this is why it is important that it is well controlled. Aside from the "smaller scale" comments provided by reviewers 2+3, the issues raised by reviewer 1 (who is an immunopeptidomics expert) are important and should be fully addressed.

Reviewer's Responses to Questions

**Part I - Summary**

Reviewer #1: While this manuscript provides some interesting data on (murine) Trypanosoma cruzi epitopes, there are major flaws which preclude publication in PLOS Pathogens. Please see section below.

Reviewer #2: The authors present a very nicely linear story that starts from square 1 and ends at an an vivo proof of concept. It is very well written and very easy (because of their explanations and writing style) to understand why the authors made all of their decisions. It is always easy to anticipate what figure/data will come next at each stage of the manuscript. Overall, a very nicely written paper.

The authors are interested in characterising the MHC-1-presented peptide profile of T cruzi infected cells, with the aim of generating a CD8 T cell focused vaccine. They use robust techniques to identify candidate vaccine targets, eventually selecting the one and only protein candidate that was detected in multiple immunopeptidomic profile experiments. I can imagine that this experiment in itself may in some cases warrant a manuscript submission, however the authors take the next step and make an attempt to pre-clinically validate this vaccine strategy. They do so by generating an mRNA vaccine delivered by LNPs and assessing its immunogenicity in a mouse model. The significance of such an effort cannot be understated, given the several million people infected with T cruzi at present and lack of any effective clinical vaccine. The novelty of the strategy is underpinned by the identification of an entirely novel vaccine target for this parasite. Although the significance of this study is undermined by the lack of some important components of your typical chagas vaccine effort, those being:

A) some stronger evidence that the vaccine will be effective across different T cruzi stains/genotypes (the protein alignments are supportive, although unconvincing. e.g., expression profile differences could play a massive role),

B) the lack of any in vivo infection studies. Given the relatively weak immune response observed in 3/5 immunized mice (addressed again in part 2), I would be surprised to learn that this vaccine (at least at its current stage of development) provides anything more than the "partielle immunité" described by Brumpt over 100 years ago after the first chagas vaccine attempt (and also observed by basically everyone else who has tried). Still, these issues do not invalidate any of the findings in the manuscript itself. And one could envision that the follow up manuscript will have several in vivo infection studies, incorporate plenty of different adjuvants, and use several different parasite strains. Incorporating all of these into this manuscript would add years worth of time and effort, which is likely beyond the intended scope of this publication.

So indeed, this manuscript is appropriately novel, is executed well, and is scholarly enough; e.g., the authors appropriately reference several previous attempts to use this same strategy to identify vaccine candidates for other pathogens. I would also mention that the figure quality is for me generally excellent. The main weaknesses of the study relate to points A and B above, in addition to some more minor weaknesses (typographical, statistics, and a few missing controls).

Reviewer #3: This manuscript by Versteeg et al. describes vaccine antigen discovery work for Chagas disease, a protozoan parasite of significant global health impact for which no vaccine currently exists. The antigen discovery work leads to identification of a specific Chagas protein (Tcj2) as a potential key cytotoxic T cell-inducing antigen. This antigen discovery work is then validated by encoding of the Tcj2 protein into an mRNA-LNP vaccine and immunization of C57Bl/6 mice, resulting in humoral and antigen-specific CD8+ T cell responses to vaccination.

There is a general lack of work in the field of vaccine development for Chagas disease, with great needs for more studies identifying potential protective antigens against this parasite such as is included in this work. This manuscript is thus novel and of significant interest to the field and to global health efforts.

The manuscript is generally well written, with various minor editorial issues. Overall this is a well thought out body of work with sound data interpretation and conclusions. The major defect of the work is the very small in vivo sample size combined with the apparent lack of any replicate in vivo study(ies). The entirety of the immune profiling of the lead identified Chagas antigen was carried out with a single study of n=5 all female C57Bl/6 mice at a single vaccine dose. The immune assays appear to be well designed and executed, and the resulting data certainly suggest support for the conclusions made, but this is insufficient to overcome the lack of in vivo sample size or any indication of technical study replicates.

**Part II – Major Issues: Key Experiments Required for Acceptance**

Reviewer #1: 1) There are no quality control measures implemented and shown for the immunopeptidomics experiment. The authors should e.g. provide figures on length distribution, binding properties, sequence clustering and retention time prediction for the identified peptides, all of which are common QC measurements in immunopeptidomics.

2) All Trypanosoma cruzi-derived peptides should be validated by comparison to synthetic peptide references.

3) The conclusion that the vaccine elicits antigen-specific T cells is almost exclusively drawn from experiments evaluating SIINFEKL specific T cells, but not T cells directed against the previously identified epitopes. Here, e.g. loadable EasyMers for murine MHCs could be used to stain peptide-specific T cell populations.

4) While Chronic Chagasic Cardiomyopathy is a human disease, all experiments in this manuscript were performed using murine model systems. The authors should strongly consider to provide evidence in the human setting by e.g. performing immunopeptidomics in Trypanosoma cruzi-infected human cell lines, using MHC-humanized mouse models (e.g. A2.DR1) and screening patient-derived PBMCs for responses against Trypanosoma cruzi epitopes identified in there.

Reviewer #2: I have three "major issues"

The authors have a very limited description of the statistics used in each figure. In the methods section, they describe only the usage of a Mann Whitney U Test. It feels to me as though this manuscript is suffering from the multiple comparisons problem. Figures 5 and 6 are particularly problematic. The authors must either:

1. strongly justify why they are not correcting for multiple comparisons, or

2. perform a post-hoc analysis on the data, or

3. repeat the in vivo immunization study to increase the robustness of the dataset.

Simply put; it's very evident that almost all of the T cell data in these two figures is driven by 1 strongly responding mouse and partially by 1 moderately mouse. This needs to be addressed somehow. It doesn't invalidate the manuscript concept, but it needs to be addressed fairly via the above options.

------

Supplementary figure 5 is devoid of any positive controls for the presence/detection of the protein/target of interest in the various comparative cell lines. I understand what the authors mean to convey here, but given the already quite weak signal in the T cruzi lane, an approx. 3-fold downregulation of the protein in human cells relative to cruzi would prevent the band from showing up. We don't know if it is even expressed at all in the comparative lysates. For me this is too problematic - this figure should be removed unless the proper positive controls can be acquired (e.g., anti-human DnaJ).

-----

Figure 2B is very weak. If this plot was represented instead as the original FACS histograms, I assume it would look like there was no change from control to experimental. The call out of Fig 2B in the text also feels somewhat out of order. And finally, the references (51 and 52) mentioned in the discussion seem to have substantially more robust datasets that suggest that MHC is downregulated by Tc infection. I think this discrepancy could easily come down to a technical / cell line issue but I think its impossible to use this (evidently not statistically significant) piece of data to make any conclusions. I would recommend either:

1. repeating this assay using a more robust technique (e.g., qPCR or a commercially reliable antibody instead of this in-house antibody), or

2. omitting this piece of data

Reviewer #3: 1. The in vivo immunogenicity study is underpowered and not replicated. This is a major drawback of the study. At a minimum a replicate study should be completed, which would at least bring the sample number to n=10 for the vaccinated mice. More appropriately, to meet current scientific standards for rigor and reproducibility, an additional study would include groups of 8-10 vaccinated mice, half male and half female, with a better control (LNP-formulated off-target mRNA) for comparison, and possibly comparing/contrasting the responses to multiple different vaccine doses rather than just one very high dose.

2. The immunopeptidomics studies are conducted using T. cruzi infected mouse fibroblast cells. Given that the ultimate goal of this vaccine development effort is to develop vaccines for human populations, why were human cells not used? Ideally human cells representing a variety of HLA class 1 haplotypes? The choice of this cell line is neither discussed nor justified, but would appear to be of substantial import for the validity of the results. This choice must be justified, and/or further data collected to validate the results (i.e., should Tcj2 peptides show up prominently as class I HLA presented peptides on human cells, in addition to the mouse data shown, that would be substantial enhancement of the key result that this may be an excellent antigen for induction of T. cruzi-reactive CTLs across species).

3. Very scant details are provided for the mRNA vaccine candidate that was engineered to express Tcj2. For example, what UTRs were used? These affect modified mRNA vaccines significantly. A statement is made that the RNA sequence was optimized for translation – how was this done, using what algorithm? What LNP system was used for encapsulation? Were there any measures of RNA integrity? It’s appreciated that basic particle size and PdI is included as part of vaccine characterization as size and charge are key parameters for cellular uptake and thus vaccine efficacy. But more information – ideally the entirety of the RNA sequence – must be included for appropriate interpretation of the data.

**Part III – Minor Issues: Editorial and Data Presentation Modifications**

Reviewer #1: (No Response)

Reviewer #2: There are a moderate number of typographical errors throughout. Some of the standouts are SIINFELK vs SIINFEKL (line 216) and the location of the SIINFEKL motif being at the 3' (the correct one according to Fig 3) vs the 5' end of the mRNA construct (line 292). The manuscript should be proofread.

The manuscript opens several lines of questions. I am unclear as to why/how the apparent majority of the T cruzi MHC-presented proteins are intracellular. The hypothesis that strikes me is based on cell death, especially since their top candidate is an HSP. Is it possible that only a small fraction of the infected cells are actually presenting these epitopes? The fraction of cells that happen to have some dead T cruzi cells within them? That may also explain the marginal (10% above control) phenotype observed in the final figure. I would recommend the authors to address this point in greater detail in the discussion.

I assume by "titer" the authors mean "end point titer" - please clarify

In some figures, e.g., Fig 2B, there is no description in the figure legend highlighting what the error bars mean. How many independent experiments? etc.

Reviewer #3: 1. Lines 174-209 discuss the similarity/conserved nature of Tcj2 proteins from related trypanosomes, concluding that the use of this antigen in vaccination may induce cross-protection against other trypanosomic infections. Much of this section is speculative and more appropriate for the Discussion section than the Results section. This reviewer recommends condensing this section in the Results and moving some content to the Discussion section.

2. Lines 194-195 state that Tcj2 and human DnaJ have 43% identity and 60% similarity, and the statement is made that this represents sufficient difference to make overlapping T cell epitopes unlikely resulting in a low risk of autoimmunity. Later statements are made about ~70% identity conservation between Tcj2 sequences in different trypanosomes being potentially sufficient for cross-protection. Clearly there is a difference between 43% and 70% similarity, but is there a known cutoff/what evidence is available to say that one is likely non-cross-reactive and the other potentially cross-reactive?

3. Discussion – the Discussion is lengthy and reads as a blow-by-blow of the Results, with a bit of additional discussion that lacks some key content. A brief discussion of the strengths and weaknesses of this study, particularly in context of the field (what does this study add to existing research) is lacking and should be added. What would the proposed final vaccine product be? Would it be focused on Tcj2 alone as an antigen, or more likely in combination with other potentially key T cruzi antigens? If so, which might be good combinations, with literature to support, and would follow-on studies be necessary to determine optimal combinations? What is known about correlates of protection against T cruzi, beyond what is briefly mentioned about IgG2? Are in vivo challenge models available that would be sufficient for such studies?

4. Statistics: why is a non-parametric test used uniformly for all statistical analyses? While nonparametric tests avoid assumptions about data normality, their use reduces statistical power relative to appropriate parametric tests. Various immune readouts – such as log-normalized serum Ab data – reliably are normally distributed and thus the use of nonparametric tests (particularly when used on an already underpowered n=5 sample size) is not necessary.

5. Tcj2 is a candidate vaccine antigen. Not a candidate vaccine. Only the Tcj2-expressing mRNA-LNP vaccine is a vaccine. Please edit the language throughout the manuscript appropriately.

6. Line 48 contains an unnecessary “a”.

7. Line 57: unclear whether the 20-30% with CCC are a subset of the 30-40% of cases that develop chronic disease, or a subset of total Chagas cases. Please clarify.

8. Line 67: the proposed vaccine would presumably prevent, rather than cure, Chagas disease, correct? If there is a therapeutic potential use for the proposed vaccine, rather than a standard prophylactic use, please make this clear as it would greatly affect the use cases for the vaccine.

9. Line 82: need to add “proteins” after MHC.

10. Line 85: “superior sensitivity” relative to what? Please clarify.

11. Lines 90-92: sentence is convoluted

12. If a major issue with the design of vaccines against Chagas is a lack of known protective epitopes, please clearly state this along with any prior knowledge in the area. Any prior knowledge is not clearly expressed in the introduction and should be.

13. Lines 97-98: Most human studies of immune responses to Covid mRNA vaccines to this reviewer’s knowledge do not indicate strong CTL responses. Please add citations for this statement.

14. Lines 98-100: The cited work here is a pre-Covid review article by many of the same authors – given the huge leap forward in understanding of immunogenicity of modified mRNA-LNP vaccines since this time, a more recent reference would be more appropriate.

15. Line 109 – Tcj2 is a candidate vaccine antigen, not a candidate vaccine.

16. Line 115 – Tcj2 expressing mRNAs formulated in LNPs.

17. Line 135 – two replicate(?) immunopeptidomics experiments. If there were any technical differences between these please clarify, and if not, state that they are technical replicates.

18. Figure 2: How many peptides that were immunoaffinity purified and made it through the LC/MS/MS process were not identifiable? If none, state so.

19. Line 190: what are TcI, TcII, and TcVI from TriTryDB? Please define or reword. If nonimportant for the conclusions, move to methods.

20. Line 195: comparisons are being made between human DnaJ and T. cruzi DnaJ-typeTcj2 genes/proteins, no? The statement as is implies that the comparisons are between the full genomes of the two rather than just these particular genes. Please reword for clarity.

21. Lines 199-200: please briefly summarize the key conclusion in reference 42 that backs this statement up. Is there a similarity threshold below which self-interactions can be safely assumed to be negligible?

22. Line 214: Tcj2-expressing mRNA vaccine.

23. While the use of a FLAG tag and SIINFEKL sequence are understandable for validation purposes, these are typically not considered to be appropriate for clinical vaccines as such tags have been known to affect expressed protein structure and stability. A brief mention, even in the Discussion, of an intent to remove these for IND-enabling studies would likely be appropriate.

24. Line 258 – what has been demonstrated is not yet the ability of Tcj2-encoding mRNAs to induce the establishment of new antigen-specific CD8+ T cell populations, but the ability of mRNA-expressed protein to trigger existing antigen-specific CD8 T cells to respond. Please reword to imply the latter rather than the former.

25. Line 258: Tcj2-expressing (or encoding) mRNAs were formulated in lipid nanoparticles (LNPs) for further in vivo assessment of Tcj2 as a vaccine antigen (or Tcj2-expressing mRNA-LNPs as a vaccine candidate). The vaccine candidate is the mRNA-LNP combination, not the RNAs alone.

26. Line 260: why were empty LNPs used as a control, rather than nonsense RNA of similar size to the Tcj2 mRNAs, or better yet, an off-target mRNA such as OVA-expressing mRNA? As noted in the data, LNPs do not form properly/uniformly if empty of RNA and may not be the most ideal control. If a follow-up or replicate in vivo study is performed, this could be easily added/changed.

27. Lines 263-265: Temperature of vaccine freezing should be mentioned (-80C?). Freezing was followed by thawing before particle size measurements by DLS, correct? If so, the term freeze/ thaw should be correctly applied.

28. Empty LNPs polydispersity index decreased significantly upon freeze/thaw? Do explain.

29. Line 272: “The PkA of LNPs has been previously shown to be a determining factor….”

30. Figure 4 panel A – column titled “LNPs” should more accurately be “LNP cargo”

31. Figure 4 panel B – there appear to be a large species (<1000 nm) present for the empty LNPs. Can you speculate what this population is and what if any affects it may have upon injection into mice?

32. Figure 5: The figure contains panels with linear and with log axes. Please define what the bars indicate for each – on log axes do they appropriately indicate geometric means and geometric stdevs? For linear axes plots, do they indicate arithmetic means and stdevs?

33. Figure 5 panel C – the rightmost bar (IgG2c serum Tcj2-binding IgG titers) shows error bars much larger than/more than spanning the displayed 4 datapoints. First – why are there just 4 datapoints rather than 5, given that there were 5 immunized mice and 5 datapoints on the other bars? Second – why are the error bars so big? Are these arithmetic stdevs rather than geometric stdevs? Please investigate for all bars in this panel.

34. Figure 5 panel E: IFNg expressing CD8+ T cell populations are surprisingly low, particularly given the typically good CD8/IFNg responses in B/6 mice. Can you explain?

35. Figure 5 flow data – please indicate on y-axes what the indicated population %s are based on. i.e., % of live CD3+CD4+ T cells, or similar.

36. Line 323 please delete or clearly define the descriptor “robust”.

37. Line 326: why was baseline titer set to 67 and why is this an appropriate LOD?

38. Lines 334-335: sentence is confusing.

39. Line 344: “…by CD8+ T cells when exposed to antigen…”

PLOS authors have the option to publish the peer review history of their article (what does this mean?). If published, this will include your full peer review and any attached files.

Reviewer #1: No

Reviewer #2: No

Reviewer #3: No
---

## [Decision Letter · Decision Letter 1]

5 Jul 2024

Dear MR Versteeg,

Thank you very much for submitting your manuscript "Immunopeptidomic MHC-I Profiling and Immunogenicity Testing Identifies Tcj2 as a New Chagas Disease mRNA Vaccine Candidate." for consideration at PLOS Pathogens. As with all papers reviewed by the journal, your manuscript was reviewed by members of the editorial board and by several independent reviewers. In light of the reviews (below this email), we would like to invite the resubmission of a significantly-revised version that takes into account the reviewers' comments.

This paper is on a very important topic (the first identification of a specific protein as a vaccine target for T cruzi (eliciting an anti-cruzi CD8+ T cell response). However, taking stock of all reviewer comments (including a reviewer who refused to review without the requested changes), it appears that the authors were not responsive to reviewer requests (which are quite reasonable and which will clearly impart robustness to the paper). Some of these comments that were not addressed at all are below:

1) immunopeptidomics was done to identify the specific epitope (but no validation with synthetic peptides was even attempted despite reviewer concerns - this is not a hard expt to do, it is a key experiment to do and it cannot be argued away)

2) vaccination to elicit T cells is reported as a single experiment of 5 mice plus 5 controls. There is variability of course, between these 5 mice. And yeah, it's 5 (five). No repeats. One can argue it's a proof of concept as the authors have done, but the counterargument is that it is a one-off. This is not robust. It may also not be reproducible. There is a reason we repeat experiments in science. This is really not negotiable.

3) The point of doing this work as a mouse expt is literally to be able to do a challenge. But, no challenge experiment was performed except ex vivo (and with middling results). Without that, the far better experiment is to do this in human (given that the disease we care about is a human disease but also, crucially, given that the epitopes are unlikely to be the same between mouse and human given HLA differences).

Given the timeliness and importance of the topic and the work, potentially, and after editorial consultation, the decision is to allow a second major revision. But this would have to incorporate the changes above (at least the 1st and 2nd point; the 3rd is not key).

We cannot make any decision about publication until we have seen the revised manuscript and your response to the reviewers' comments. Your revised manuscript is also likely to be sent to reviewers for further evaluation.

Sincerely,

F. Nina Papavasiliou

Academic Editor

PLOS Pathogens

Jeffrey Dvorin

Section Editor

PLOS Pathogens

Michael Malim

Editor-in-Chief

PLOS Pathogens

orcid.org/0000-0002-7699-2064

This paper is on a very important topic (the first identification of a specific protein as a vaccine target for T cruzi (eliciting an anti-cruzi CD8+ T cell response). However, taking stock of all reviewer comments (including a reviewer who refused to review without the requested changes), it appears that the authors were not responsive to reviewer requests (which are quite reasonable and which will clearly impart robustness to the paper). Some of these comments that were not addressed at all are below:

1) immunopeptidomics was done to identify the specific epitope (but no validation with synthetic peptides was even attempted despite reviewer concerns - this is not a hard expt to do, it is a key experiment to do and it cannot be argued away)

2) vaccination to elicit T cells is reported as a single experiment of 5 mice plus 5 controls. There is variability of course, between these 5 mice. And yeah, it's 5 (five). No repeats. One can argue it's a proof of concept as the authors have done, but the counterargument is that it is a one-off. This is not robust. It may also not be reproducible. There is a reason we repeat experiments in science. This is really not negotiable.

3) The point of doing this work as a mouse expt is literally to be able to do a challenge. But, no challenge experiment was performed except ex vivo (and with middling results). Without that, the far better experiment is to do this in human (given that the disease we care about is a human disease but also, crucially, given that the epitopes are unlikely to be the same between mouse and human given HLA differences).

Given the timeliness and importance of the topic and the work, potentially, and after editorial consultation, the decision is to allow a second major revision. But this would have to incorporate the changes above (at least the 1st and 2nd point; the 3rd is not key).

Reviewer's Responses to Questions

**Part I - Summary**

Reviewer #2: I feel that the authors have done a reasonable job trying to satisfy the majority of the reviewer comments. It is certainly notable that they have elected not to pursue the majority of the experimental suggestions, although I personally feel that the manuscript is robust enough and they have certainly addressed my review comments thoughtfully and completely.

Reviewer #3: The strengths and weaknesses of this study have not significantly changed in this revision. Strengths include generally well executed tackling of vaccine antigen discovery for Chagas disease, a neglected disease and worthy target of such work. The weaknesses remain the conduct of the vaccine antigen discovery work using non-human cells (though this is now much better acknowledged in the revised manuscript), and the very low animal numbers in the single in vivo conducted experiment.

**Part II – Major Issues: Key Experiments Required for Acceptance**

Reviewer #2: (No Response)

Reviewer #3: (No Response)

**Part III – Minor Issues: Editorial and Data Presentation Modifications**

Reviewer #2: (No Response)

Reviewer #3: The main remaining issue is the low mouse numbers in the single in vivo study. As the authors acknowledge, there is presumably funding or other limitations preventing them from replicating the singular small mouse immunogenicity study presented. The work is valid as-is and worthy of publication in this reviewer's judgment - I suspect that the data would repeat well - but this study remains significantly underpowered and without replication. This reviewer leaves it in the editor's ultimate hands as to whether a single study with n=5 mice is sufficient scientific rigor for this particular journal.

Other minor issues:

Line 408-409 of the revised manuscript now contain a sentence fragment.

Additional statistical information is appreciated. The use of nonparametric tests is still not necessary, but is not invalid and thus acceptable. The multiple testing corrections mentioned by another reviewer are unnecessary as only two groups are being tested in any one test.

The addition of further information about the mRNA used (i.e., what the UTRs are derived from) is appreciated - and should be added to the Methods section of the paper.

PLOS authors have the option to publish the peer review history of their article (what does this mean?). If published, this will include your full peer review and any attached files.

Reviewer #2: No

Reviewer #3: No
---

## [Editor Report · Decision Letter 2]

19 Nov 2024

Dear MR Versteeg,

Thanks very much for revising your manuscript 'Immunopeptidomic MHC-I Profiling and Immunogenicity Testing Identifies Tcj2 as a New Chagas Disease mRNA Vaccine Candidate.' according to reviewer requests. I am pleased tp inform you that it has been provisionally accepted for publication in PLOS Pathogens.

Best regards,

F. Nina Papavasiliou

Academic Editor

PLOS Pathogens

Jeffrey Dvorin

Section Editor

PLOS Pathogens

Michael Malim

Editor-in-Chief

PLOS Pathogens

orcid.org/0000-0002-7699-2064
---

## [Editor Report · Acceptance letter]

2 Dec 2024

Dear MR Versteeg,

We are delighted to inform you that your manuscript, "Immunopeptidomic MHC-I Profiling and Immunogenicity Testing Identifies Tcj2 as a New Chagas Disease mRNA Vaccine Candidate.," has been formally accepted for publication in PLOS Pathogens.

Best regards,

Michael Malim

Editor-in-Chief

PLOS Pathogens

orcid.org/0000-0002-7699-2064